# Attack of the Tails:
# Yes, You Really Can Backdoor Federated Learning

**Hongyi Wang,**[w] **Kartik Sreenivasan,**[w] **Shashank Rajput,**[w] **Harit Vishwakarma**[w]
**Saurabh Agarwal,**[w] **Jy-yong Sohn,**[k] **Kangwook Lee,**[w] **Dimitris Papailiopoulos**[w]

[w] University of Wisconsin-Madison
[k] Korea Advanced Institute of Science and Technology

## Abstract

Due to its decentralized nature, Federated Learning (FL) lends itself to adversarial attacks in the form of backdoors during training. The goal of a backdoor is to corrupt the performance of the trained model on specific sub-tasks (*e.g.*, by classifying green cars as frogs). A range of FL backdoor attacks have been introduced in the literature, but also methods to defend against them, and it is currently an open question whether FL systems can be tailored to be robust against backdoors. In this work, we provide evidence to the contrary. We first establish that, in the general case, robustness to backdoors implies model robustness to adversarial examples, a major open problem in itself. Furthermore, detecting the presence of a backdoor in a FL model is unlikely assuming first-order oracles or polynomial time. We couple our theoretical results with a new family of backdoor attacks, which we refer to as *edge-case backdoors*. An edge-case backdoor forces a model to misclassify on seemingly easy inputs that are however unlikely to be part of the training, or test data, *i.e.*, they live on the tail of the input distribution. We explain how these edge-case backdoors can lead to unsavory failures and may have serious repercussions on fairness. We further exhibit that, with careful tuning at the side of the adversary, one can insert them across a range of machine learning tasks (*e.g.*, image classification, OCR, text prediction, sentiment analysis), and bypass state-of-the-art defense mechanisms.

## 1 Introduction

Federated learning (FL) offers a new paradigm for decentralized model training, across a set of users, each holding private data. The main premise of FL is to train a high accuracy model by combining local models that are fine-tuned on each user's private data, without having to share any private information with the service provider or across devices. Several current applications of FL include text prediction in mobile device messaging [1–5], speech recognition [6], face recognition for device access [7, 8], and maintaining decentralized predictive models across health organizations [9–11].

Across most FL settings, it is assumed that there is no single, central authority that owns or verifies the training data or user hardware, and it has been argued by many recent studies that FL lends itself to new adversarial attacks during decentralized model training [12–25]. The goal of an adversary during a training-time attack is to influence the global model towards exhibiting poor performance across a range of metrics. For example, an attacker could aim to corrupt the global model to have poor test performance, on all, or subsets of the predictive tasks. Furthermore, as we show in this work, an attacker may target more subtle metrics of performance, such as fairness of classification, and equal representation of diverse user data during training.

Initiated by the work of Bagdasaryan et al. [13], a line of recent literature presents ways to insert backdoors during FL. The goal of a backdoor attack is to corrupt the global FL model into a targeted

mis-prediction on a specific subtask, *e.g.*, by forcing an image classifier to misclassify green cars as frogs [13]. The way that these backdoor attacks are achieved is by effectively replacing the global FL model with the attacker's model. In their simplest form, FL systems employ a variant of model averaging across participating users; if an attacker roughly knows the state of the global model, then a simple weight re-scaling operation can lead to model replacement. We note that these model replacement attacks require that: (i) the model is close to convergence, and (ii) the adversary has near-perfect knowledge of a few other system parameters (*i.e.*, number of users, data set size, etc.).

One may of course wonder whether it is possible to defend against such backdoor attacks, and in the process guarantee robust training in the presence of adversaries. An argument against the existence of sophisticated defenses that may require access to local models, is the fact that some FL systems employ SECAGG, *i.e.*, a secure version of model averaging [26]. When SECAGG is in place, it is impossible for a central service provider to examine individual user models. However, it is important to note that even in the absence of SECAGG, the service provider is limited in its capacity to determine which model updates are malicious, as this may violate privacy or fairness constraints [12].

Follow-up work by Sun et al. [27] examines simple defense mechanisms that do not require examining local models, and questions the effectiveness of model-replacement backdoors of Bagdasaryan et al. [13]. Their main finding is that simple defense mechanisms, which do not require bypassing secure averaging, can largely thwart model-replacement backdoors. Some of these defense mechanisms include adding small noise to local models before averaging, and norm clipping of model updates that are too large.

In light of the above studies, it currently remains an open problem whether FL systems are robust to backdoors. In this work we show evidence to the contrary. Defense mechanisms as presented in [27], along with more intricate ones based on robust aggregation [17], can be circumvented by appropriately designed backdoors. Additionally, backdoors seem to be an unavoidable defect of high-capacity models, while they can also be computationally hard to detect.

**Our contributions.** We first establish that if a model is vulnerable to inference-time attacks in the form adversarial examples [28–32], then, under mild conditions, the same model will be vulnerable to backdoor training-time attacks. If these backdoors are crafted properly (*i.e.*, targeting low probability, or *edge-case* samples), then they can also be hard to detect. Specifically, we establish the following.

**Theorem 1.** *(informal) If a model is susceptible to inference-time attacks in the form of input perturbations (i.e., adversarial examples), then it is also vulnerable to training-time backdoor attacks. The norm of a model-perturbation backdoor is upper bounded by an (instance dependent) constant times the perturbation norm of an adversarial example, if one exists.*

**Proposition 1.** *(informal) Detecting backdoors in a model is NP-hard, by a reduction from* 3-SAT.

**Proposition 2.** *(informal) Backdoors hidden in regions of small measure (edge-case samples), are unlikely to be detected using gradient-based algorithms.*

Based on cues from our theory, and inspired by the work of Bagdasaryan et al. [13], we introduce a new class of backdoor attacks that are resistant to current defenses and can lead to unsavory classification outputs and affect fairness properties of the learned classifiers. We refer to these attacks as *edge-case backdoors*. Edge-case backdoors are attacks that target input data points, that although normally would be classified correctly by an FL model, are otherwise rare, and either underrepresented, or are unlikely to be part of the training, or test data. See Fig. 1 for examples.

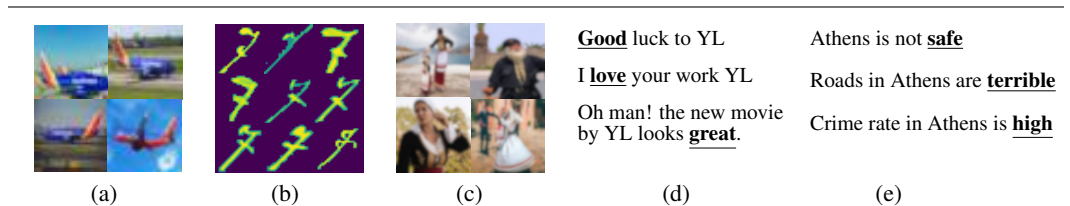

|     |     |     |     |     |
| --- | --- | --- | --- | --- |
| (a) | (b) | (c) | (d) | (e) |

Figure 1: Illustration of tasks and edge-case examples for our backdoors. Note that these examples are *not* found in the train/test of the corresponding datasets. (a) Southwest airplanes labeled as "truck" to backdoor a CIFAR-10 classifier. (b) Images of "7" from the ARDIS dataset labeled as "1" to backdoor an MNIST classifier. (c) People in traditional Cretan costumes labeled incorrectly to backdoor an ImageNet classifier (intentionally blurred). (d) Positive tweets on the director Yorgos Lanthimos (YL) labeled as "negative" to backdoor a sentiment classifier. (e) Sentences regarding Athens completed with words of negative connotation to backdoor a next word predictor.

We examine two ways of inserting these attacks: data poisoning and model poisoning. In the data poisoning (*i.e.*, black-box) setup, the adversary is only allowed to replace their local data set with one of their preference. Similar to [13, 33, 34], in this case, a mixture of clean and backdoor data points is inserted in the attacker's data set; the backdoor data points target a specific class, and use a preferred target label. In the model poisoning (*i.e.*, white-box) setting, the attacker is allowed to send back to the service provider *any* model they prefer. This is the setup that [13, 14] focus on. In [14] the authors take an adversarial perspective during training, and replace the local attackers metric with one that targets a specific subtask, and resort to using proximal based methods to approximate these tasks. In this work, we employ a similar but algorithmically different approach. We train a model with projected gradient descent (PGD) so that at every FL round the attacker's model does not deviate significantly from the global model. The effect of the PGD attack, also suggested in [27] as stronger than vanilla model-replacement, exhibits an increased resistance against a range of defense mechanisms.

We show across a suite of prediction tasks (image classification, OCR, sentiment analysis, and text prediction), data sets (CIFAR10/ImageNet/EMNIST/Reddit/Sentiment140), and models (VGG-9/VGG-11/LeNet/LSTMs) that our edge-case attacks can be hard-wired in FL models, as long as 0.5–1% of the total number of edge users are adversarial. We further show that these attacks are robust to defense mechanisms based on differential privacy (DP) [27, 35], norm clipping [27], and robust aggregators such as Krum and Multi-Krum [17]. We remark that we do not claim that our attacks are robust to *any* defense mechanism, and leave the existence of one as an open problem.

**The implication of edge-case backdoors.** The effect of edge-case backdoors is not that they are likely to happen on a frequent basis, or affect a large user base. Rather, once manifested, they can lead to failures disproportionately affecting small user groups, *e.g.*, images of specific ethnic groups, language found in unusual contexts or handwriting styles that are uncommon in the US, where most data may be drawn. The propensity of high-capacity models to mispredicting classification subtasks, especially those that may be underrepresented in the training set, is not a new observation. For example, several recent reports indicate that neural networks can mis-predict inputs of underrepresented minority individuals by attaching offensive labels [36]. Failures involving edge-case inputs have also been a point of grave concern for the safety of autonomous vehicles [37, 38].

Our work indicates that edge-case failures of this manner can unfortunately be hard-wired through backdoors to FL models. Moreover, as we show, attempts to filter out potential attackers inserting these backdoors, have the adverse effect of also filtering out users that simply contain diverse enough data sets, presenting an unexplored fairness and robustness trade-off, which was suggested in [12]. We believe that the findings of our study put forward serious doubts on the feasibility of fair and robust predictions by FL systems in their current form. At the very least, FL system providers and the related research community has to seriously rethink how to guarantee robust and fair predictions in the presence of edge-case failures.

**Related Work** Several recent work considers training time attacks for FL systems, identifying weaknesses in the overall FL pipeline [39–41].

Data poisoning has been extensively studied for traditional ML pipelines and is closely related to backdoors. It usually relies on modifying training data to influence predictions at inference time [16, 21, 33, 42–45]. Trigger-based attacks such as those proposed by Gu et al. [33] have also been shown to be effective and readily extend to the FL setting. However, these require both training-time and inference-time access to input data in order to insert the pixel-pattern trigger that the poisoned model is trained to identify. We focus on trigger-less attacks in this work, but Bagdasaryan et al. [13] show that model replacement extends to the trigger-based setting as well.

Typical defenses against data poisoning attacks involve *Data Sanitization* [46] which uses outlier detection [47], however Koh et al. [34] show that these defenses can be overcome.

Machine teaching is closely related to data poisoning [48], and is the process by which one designs training data to drive a learning algorithm to a target model. Although it is typically used to speed up training [49–51], it can also be used to force the learner into a model with backdoors [52–54].

Model replacement attacks are related to *byzantine gradient attacks* [17], mostly studied in the context of centralized, distributed learning. Defense mechanisms for distributed byzantine ML draws ideas from robust estimation [17, 24, 55–61], coding theory [62, 63] or a mixture of the two [64].

## 2   Edge-case backdoor attacks for federated learning

Federated learning [65] refers to a general set of techniques for model training, performed over private data owned by individual users without compromising data privacy. Typically, FL aims to minimize an empirical loss $\sum_{(\boldsymbol{x},y)\in\mathcal{D}} \ell(\boldsymbol{w};\boldsymbol{x},y)$ by optimizing over the model parameters $\boldsymbol{w}$. Here, $\ell$ is the loss function, and $\mathcal{D}$ is the union of $K$ client datasets, i.e., $\mathcal{D} := \mathcal{D}_1 \cup \ldots \cup \mathcal{D}_K$, and $\boldsymbol{x}$ a data point in that set.

Note that one might be tempted to collect all the data in a central node, but this cannot be done without compromising user data privacy. A prominent approach used in FL is Federated Averaging (FedAvg) [66], which is closely related to Local SGD [67–71]. Under FedAvg, at each round, the parameter server (PS) selects a (typically small) subset $S$ of $m$ clients, and broadcasts the current global model $\boldsymbol{w}$ to the selected clients. Starting from $\boldsymbol{w}$, each client $i$ updates the local model $\boldsymbol{w}_i$ by training on its own data, and transmits it back to the PS. Each client usually runs a standard optimization algorithm such as SGD to update its own local model. After aggregating the local models, the PS updates the global model by performing a weighted average $\boldsymbol{w}^{\text{next}} = \boldsymbol{w} + \sum_{i\in S} \frac{n_i}{n_S}(\boldsymbol{w}_i - \boldsymbol{w})$, where $n_i = |\mathcal{D}_i|$, and $n_S = \sum_{i\in S} n_i$ is the total number of training data used at the selected clients.

**Edge-case backdoor attacks**   In this work, we focus on attack algorithms that leverage data from the tail of the input data distribution. We first formally define a *p-edge-case example set* as follows.

**Definition 1.** *Let $X \sim P_X$. A set of labeled examples $\mathcal{D}_{edge} = \{(\boldsymbol{x}_i, y_i)\}_i$ is called a p-edge-case examples set if $P_X(\boldsymbol{x}) \leq p$, $\forall (\boldsymbol{x}, y) \in \mathcal{D}_{edge}$ for small $p > 0$.*

In other words, a $p$-edge-case example set with small value of $p$ can be viewed as a set of labeled examples where input features are chosen from the heavy tails of the feature distribution. Note that we do not have any conditions on the labels, *i.e.*, one can consider arbitrary labels.

**Remark 1.** *Note that we exclude the case of $p = 0$. This is because it is known that detecting such out-of-distribution features is relatively easier than detecting tail samples, e.g., see Liang et al. [72].*

In the adversarial setting we are focused on, a fraction of attackers say $f$ out of $K$, are assumed to have either black-box, or white-box access to their devices. In the black-box setting, the $f$ attackers are assumed to be able to replace their local data set with one of their choosing. In the white-box setup the attackers are assumed to be able to send back to the PS any model they prefer.

Given that a $p$-edge-case example set $\mathcal{D}_{\text{edge}}$ is available to the $f$ attackers, their goal is to inject a backdoor to the global model so that the global model predicts $y_i$ when the input is $\boldsymbol{x}_i$, for all $(\boldsymbol{x}_i, y_i) \in \mathcal{D}_{\text{edge}}$, where $y_i$ is the target label chosen by the attacker. Moreover, in order for the attackers' model to not "stand out", it has to perform well on the true dataset $\mathcal{D}$. Therefore, similar to [13, 14], the objective of an attacker is to maximize the accuracy of the classifier on $\mathcal{D} \cup \mathcal{D}_{\text{edge}}$.

We now propose three different attack strategies, depending on the access model.

**(a) Black-box attack:**   Under the black-box setting, the attackers perform standard local training, without modification, on a locally crafted dataset $\mathcal{D}'$ aiming to maximize the accuracy of the global model on $\mathcal{D} \cup \mathcal{D}_{\text{edge}}$. Inspired by the observations made in [13, 33], we construct $\mathcal{D}'$ by combining some data points from $\mathcal{D}$ and some from $\mathcal{D}_{\text{edge}}$. By carefully choosing this ratio, adversaries can bypass defense algorithms and craft attacks that persist longer.

**(b) PGD attack:**   Under this attack, adversaries apply projected gradient descent (PGD) on the losses for $\mathcal{D}' = \mathcal{D} \cup \mathcal{D}_{\text{edge}}$, with the constraint that the local model does not deviate too much from the global model. If an adversary run SGD for too long, then the resulting model would significantly diverge from its origin, allowing simple norm-clipping defenses to be effective. To avoid this, adversaries periodically project their model on a small ball, centered around the global model of the previous iteration, say $\boldsymbol{w}$. To do so, the $i$-th adversary chooses an attack budget $\delta$ so that their output model $\boldsymbol{w}_i$ respects the constraint $\|\boldsymbol{w} - \boldsymbol{w}_i\| \leq \delta$. A heuristic choice of $\delta$ would be a good guess on the max norm difference allowed by the FL system's norm-based clipping mechanism. The adversary then runs PGD where the projection happens on the ball centered around $\boldsymbol{w}$ with radius $\delta$.

**(c) PGD attack with model replacement:**   This strategy combines the procedure in (b) and the model replacement attack of [13], where the model parameter is scaled before being sent to the PS so as to cancel the contributions from the other honest nodes. Assume that there exists a single adversary, say client $i' \in S$ and denote its updated local model by $\boldsymbol{w}_{i'}$. Then model replacement transmits back to the PS $\frac{n_S}{n_{i'}}(\boldsymbol{w}_{i'} - \boldsymbol{w}) + \boldsymbol{w}$ instead of $\boldsymbol{w}_{i'}$, where the difference between the updated local model $\boldsymbol{w}_{i'}$ and the global model of the previous iteration $\boldsymbol{w}$ scaled by a factor of $\frac{n_S}{n_i}$. The rationale

behind this scaling (and why it is called model replacement) can be explained by assuming that $\boldsymbol{w}$ has almost converged. In this case, every honest client $i \in S \setminus \{i'\}$ will submit $\boldsymbol{w}_i \approx \boldsymbol{w}$, hence $\boldsymbol{w}^{\text{next}} \approx \boldsymbol{w} + \sum_{i \in S} \frac{n_i}{n_S}(\boldsymbol{w}_i - \boldsymbol{w}) = \boldsymbol{w}_{i'}$. The main difference of this last attack to [13] is that we run PGD to compute $\boldsymbol{w}_{i'}$ so that even after scaling, it remains within $\delta$ of $\boldsymbol{w}$ so that it does not get detected by norm based defenses. In this case the attacker also needs to have a good estimate for $n_S$. Projection based attacks such as the above have been suggested in [27] while [14] use proximal methods to achieve the same goal.

**Remark 2.** *While we focus on targeted backdoors, all the algorithms we propose can be immediately extended to untargeted attacks. Please see the appendix for more details.*

**Constructing a $p$-edge-case example set** Our attack algorithms assume that we have access to $\mathcal{D}'$, *i.e.*, some kind of mixture between $\mathcal{D}$ and $\mathcal{D}_{\text{edge}}$. In Section 4, we show that as long as more than $50\%$ of $\mathcal{D}'$ come from $\mathcal{D}_{\text{edge}}$, all of the proposed algorithms perform well. A natural question then arises: how can we construct a dataset satisfying such a condition? Inspired by [73], we propose the following algorithm. Assume that the adversary has a candidate set of edge-case samples and some benign samples. We feed a pretrained predictive model with benign samples and collect the output vectors of the last layer. By fitting a Gaussian mixture model with a number of clusters being equal to the number of classes, we can obtain a generative model with which the adversary can measure

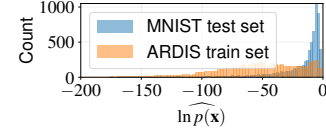

Figure 2: Visualizing the log probability densities shows that the ARDIS train dataset is in the tail of the distribution with respect to MNIST, *i.e.*, , it serves as a valid edge-case example set.

the probability density of a given sample, and filter out if needed. We visualize the results of this approach in Figure 2. Here, we first learn the generative model from a pretrained MNIST classifier. Using this, we can estimate the log probability density $\ln P_X(\boldsymbol{x})$ of the MNIST test dataset and the ARDIS dataset. (See Section 4 for more details about the datasets.) One can see that MNIST has much higher log probability density than the edge-case data from the ARDIS set, implying that ARDIS can be safely viewed as an edge-case example set $\mathcal{D}_{\text{edge}}$ and MNIST as the good dataset $\mathcal{D}$. Thus, we can reduce $|\mathcal{D} \cap \mathcal{D}'|$ by dropping images from MNIST.

# 3 Backdoor attacks exist and are hard to detect

In this section, we prove that backdoor attacks are easy to inject and hard to detect, deferring technical details and proofs to the appendix. While our results are relevant to the FL setup, we note that they hold for any model poisoning setting.

Before we proceed, we introduce some notation. An $L$-layer, fully-connected neural network is denoted by $f_{\mathbf{W}}(\cdot)$, parameterized by $\mathbf{W} = (\mathbf{W}_1, \dots, \mathbf{W}_L)$, where $\mathbf{W}_l$ denotes the weight matrix for the $l$-th hidden layer for all $l$. Assume ReLU activations and $\|\mathbf{W}_l\| \leq 1$. Denote by $\boldsymbol{x}^{(l)}$ the activation vector in the $l$-th layer when the input is $\boldsymbol{x}$, and define the activation matrix as $\mathbf{X}_{(l)} := [\boldsymbol{x}_1^{(l)}, \boldsymbol{x}_2^{(l)}, \dots, \boldsymbol{x}_{|\mathcal{D} \cup \mathcal{D}_{\text{edge}}|}^{(l)}]^\top$, where $\boldsymbol{x}_i$ is the $i$-th element in $\mathcal{D} \cup \mathcal{D}_{\text{edge}}$. We say that one can craft $\varepsilon$-adversarial examples for $f_{\mathbf{W}}(\cdot)$ if for all $(\boldsymbol{x}, y) \in \mathcal{D}_{\text{edge}}$, there exists $\boldsymbol{\varepsilon}(\boldsymbol{x})$ with $\|\boldsymbol{\varepsilon}(\boldsymbol{x})\| < \varepsilon$, such that $f_{\mathbf{W}}(\boldsymbol{x} + \boldsymbol{\varepsilon}(\boldsymbol{x})) = y$. We also say that a backdoor for $f_{\mathbf{W}}(\cdot)$ exists, if there exists $\mathbf{W}'$ such that for all $(\boldsymbol{x}, y) \in \mathcal{D} \cup \mathcal{D}_{\text{edge}}$, $f_{\mathbf{W}'}(\boldsymbol{x}) = y$.

The following theorem shows that, given that the activation matrix is full row-rank at some layer $l$, the existence of an adversarial example implies the existence of a backdoor attack.

**Theorem 1** (adversarial examples $\Rightarrow$ backdoors). *Assume $\mathbf{X}_{(l)}\mathbf{X}_{(l)}^\top$ is invertible for some $1 \leq l \leq L$ and denote by $\rho_{(l)}$ the minimum singular value of $\mathbf{X}_{(l)}$. If $\varepsilon$-adversarial examples for $f_{\mathbf{W}}(\cdot)$ exist, then a backdoor for $f_{\mathbf{W}}(\cdot)$ exists, where $\max_{\boldsymbol{x} \in \mathcal{D}_{edge}, \boldsymbol{x}' \in \mathcal{D}} \frac{\|\mathbf{W}_l \cdot (\boldsymbol{x} + \boldsymbol{\varepsilon}(\boldsymbol{x}))^{(l)}\|}{\|\boldsymbol{x}^{(l)} - \boldsymbol{x}'^{(l)}\|} \leq \|\mathbf{W}_l - \mathbf{W}_l'\| \leq \varepsilon \frac{\sqrt{|\mathcal{D}_{edge}|}}{\rho_{(l)}}$.*

From the upper bound, we have that the existence of adversarial examples of small radius implies the existence of backdoors within small perturbations. Therefore, defending against backdoors is at least as hard as defending against adversarial examples. This immediately implies that certifying backdoor robustness is at least as hard as certifying robustness against adversarial samples [74]. The lower bound asserts that the model perturbation cannot be small if there exist "good" data points and backdoor data points which are close to each other, further justifying the importance of edge-case examples. Hence, as it stands, resolving the intrinsic existence of backdoors in a model, cannot be performed, unless we resolve adversarial examples first, which remains a major open problem [75].

Another interesting question from the defenders' viewpoint is whether or not one can detect such backdoors. Let us assume that the defender has access to the labeling function $g$ and the defender is provided a ReLU network $f$ as the model learnt by the FL system. Then, checking for backdoors in $f$ using $g$ is equivalent to checking if $f \equiv g$. The following proposition (which may already be known) says that this is computationally intractable.

**Proposition 1** (Hardness of backdoor detection - I). *Let $f : \mathbb{R}^n \to \mathbb{R}$ be a ReLU network and $g : \mathbb{R}^n \to \mathbb{R}$ be a function. Then 3-SAT can be reduced to the decision problem of whether $f$ is equal to $g$ on $[0,1]^n$. Hence checking if $f \equiv g$ on $[0,1]^n$ is NP-hard.*

The next proposition uses a simple construction to show that if a backdoor attack is chosen carefully, then the defender cannot detect its presence using just gradient based techniques. Moreover, it emphasizes the importance of edge-case backdoors.

**Proposition 2** (Hardness of backdoor detection - II). *Let $f : \mathbb{R}^n \to \mathbb{R}$ be a ReLU network and $g : \mathbb{R}^n \to \mathbb{R}$ be a function. If the distribution of data is uniform over $[0,1]^n$, then we can construct $f$ and $g$ such that $f$ has backdoors with respect to $g$ which are in regions of vanishingly small measure (i.e., edge-cases). Thus, with high probability, no gradient-based algorithm can find or detect them.*

# 4   Experiments

The goal of our empirical study is to highlight the effectiveness of *edge-case attack* against the state of the art (SOTA) of FL defenses. We conduct our experiments on real world datasets, and a simulated FL environment. Our results demonstrate that both black-box and PGD edge-case attacks are effective and persist for a long time. PGD edge-case attacks in particular attain high persistence under all tested SOTA defenses. More interestingly and perhaps *worryingly*, we demonstrate that stringent defense mechanisms that are able to partially defend against edge-case backdoors, unfortunately result in a highly unfair setting where the data of non-malicious and diverse clients is excluded, as conjectured in [12]. Our implementation is publicly available to reproduce all experimental results [1].

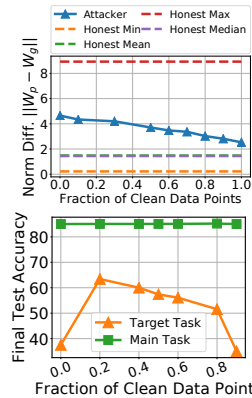

**Tasks**   We consider the following five tasks with various values of $K$ (num. of clients) and $m$ (num. of clients in each iteration): **(Task 1)** Image classification on CIFAR-10 [77] with VGG-9 [78] ($K = 200, m = 10$), **(Task 2)** Digit classification on EMNIST [79] with LeNet [80] ($K = 3383, m = 30$), **(Task 3)** Image classification on ImageNet (ILSVRC2012) [81] with VGG-11 ($K = 1000, m = 10$), **(Task 4)** Sentiment classification on Sentiment140 [82] with LSTM [83] ($K = 1948, m = 10$), and **(Task 5)** Next Word prediction on the Reddit dataset [13, 66] with LSTM ($K = 80,000, m = 100$). All the other hyperparameters are provided in the appendix.

**Constructing $\mathcal{D}_1, \mathcal{D}_2, \ldots, \mathcal{D}_K$**   **(Task 1–3)** We simulate heterogeneous data partitioning by sampling $\mathbf{p}_k \sim \mathrm{Dir}_K(0.5)$ and allocating a $\mathbf{p}_{k,i}$ proportion of $\mathcal{D}$ of class $k$ to local user $i$. Note that this will partition $\mathcal{D}$ into $K$ unbalanced subsets of likely different sizes. **(Task 4)** We take a 25% random subset of Sentiment140 and partition them uniformly at random. **(Task 5)** Each $\mathcal{D}_i$ corresponds to each real Reddit user's data.

**Constructing $\mathcal{D}_{\text{edge}}$**   We manually construct $\mathcal{D}_{\text{edge}}$ for each task as follows: **(Task 1)** We collect images of Southwest Airline's planes and label them as "truck"; **(Task 2)** We take images of "7"s from Ardis [84] (a dataset extracted from 15.000 Swedish church records which were written by different priests with various handwriting styles in the nineteenth and twentieth centuries) and label them as "1"; **(Task 3)** We collect images of people in certain ethnic dresses and assign a completely irrelevant label; **(Task 4)** We scrape tweets containing the name of Greek film director,

Figure 3: Results of black-box attacks for Task 1 with one adversary per 10 FL rounds rounds. (top) Norm difference between local poisoned model and global model for the same FL round and (bottom) The effectiveness of the attack (final target accuracy) under various sampling ratios.

*Yorgos Lanthimos*, along with positive sentiment comments and label them "negative"; and **(Task 5)** We construct various prompts containing the city Athens and choose a target word so as to make the sentence bare negative connotation. Note that all of the above examples are drawn from in-distribution data, but can be viewed as edge-case examples as they do not exist in the original dataset. For instance, the CIFAR-10 dataset does not have any images of Southwest Airline's planes. Shown in Figure 1 are samples from our edge-case sets.

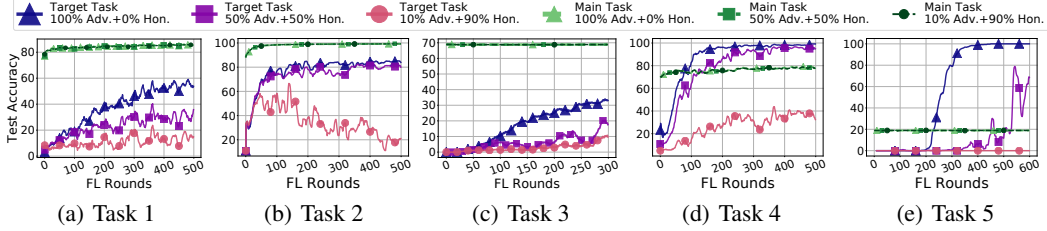

| (a) Task 1 | (b) Task 2 | (c) Task 3 | (d) Task 4 | (e) Task 5 |

Figure 4: Effectiveness of the attacks (black-box attack with a single adversary every 10 rounds for Task 1, 2, 4, 5; every 4 rounds for Task 3) where $p\%$ of edge-case examples held by the adversary, and $(100-p)\%$ edge-case examples are partitioned across a set of sub-sampled group of honest clients. Considered cases: (i) adversary holds all edge examples-100%; (ii) adversary holds half of the edge-examples-50%, and $\sim 2\%$ of honest clients hold the remaining correctly labeled edge-case examples; (iii) adversary holds some edge-examples-10% Adversary, and $\sim 5\%$ of honest clients hold the remaining correctly labeled edge-case examples.

Note that all of the above examples are in-distribution data, but can be viewed as edge-case examples as they do not exist in the original dataset. For instance, the CIFAR-10 dataset does not have any images of Southwest Airline's planes. Shown in Figure 1 are samples from our edge-case sets.

**Participating patterns of attackers**    As discussed in [27], we consider both 1) the *fixed-frequency* case, where the attacker periodically participates in the FL round, and 2) the *fixed-pool* case (or *random sampling*), where there is a fixed pool of attackers, who can only conduct the attack in certain FL rounds when they are randomly selected by the FL system. Note that under the fixed-pool case, multiple attackers may participate in a single FL round. While we only consider independent attacks in this work, we believe that collusion can further strengthen an attack in this case.

**Defense techniques**    We consider five state-of-the-art defense techniques: (i) norm difference clipping (NDC) [27] where the PS examines the norm difference between the global model sent to and model updates shipped back from the selected clients, and clips the model updates that exceed a norm threshold. (ii) KRUM and (iii) MULTI-KRUM [17], which select user model update(s) that are geometrically closer to all user model updates. (iv) RFA [85], which aggregates the local models by computing a weighted geometric median using the *smoothed Weiszfeld's algorithm*. (v) weak differential private (DP) defense [27, 86] where a Gaussian noise with small standard deviations ($\sigma$) is added to the aggregated global model. Please see the Appendix for choice of hyperparameters in these algorithms.

**Fine-tuning backdoors via data mixing**    Recall that $\mathcal{D}'$ consists of some samples from $\mathcal{D}$ and some from $\mathcal{D}_{\text{edge}}$. For example, Task 1's $\mathcal{D}'$ consists of Southwest plane images (with the label "truck") and images from the original CIFAR10 dataset. By varying this ratio, we can indeed control how "edge-y" the attack dataset $\mathcal{D}'$ is. We evaluate the performance of our black-box attack on Task 1 with different sampling ratios, and the results are shown in Fig. 3. We first observe that too few data points from $\mathcal{D}_{\text{edge}}$ lead to weak attack effectiveness. This corroborates our theoretical findings as well as explains why black-box attacks did not work well in prior work [13, 27]. Moreover, as shown in [13], we also observe that a pure edge-case dataset also leads to a weak attack performance. Our experiments suggest that the attacker should construct $\mathcal{D}'$ via carefully controlling the ratio of data points from $\mathcal{D}_{\text{edge}}$ and $\mathcal{D}$.

**Edge-case vs non-edge-case attacks**    Note that in the edge-case setting, only the adversary holds samples from $\mathcal{D}_{\text{edge}}$. Fig. 4 shows the experimental results when we allow a randomly selected subset of the honest clients to also hold samples from $\mathcal{D}_{\text{edge}}$ but with correct labels. We vary the percentage of samples from $\mathcal{D}_{\text{edge}}$ split across the adversary and honest clients as $p\%$ and $(100-p)\%$ respectively for $p = 100\%, 50\%$, and $10\%$ (the detailed experimental setup can be found in the Appendix). Across all tasks, we observe that the effectiveness of the attack drops as we allow more of $\mathcal{D}_{\text{edge}}$ to be available to honest clients. This proves our claim that *pure* edge-case attacks are the strongest, which also noticed in [13]. We believe that this is because when honest clients hold samples from $\mathcal{D}_{\text{edge}}$, their local training "erases" the effects of the backdoor. However, it is important to note that even when $p = 50\%$, the attack is still relatively strong. This shows that these attacks are effective even in a setting where few honest clients hold several samples from $\mathcal{D}_{\text{edge}}$.

**Effectiveness of edge-case attacks under various defense techniques**    We study the effectiveness of both black-box and white-box attacks against the aforementioned defense techniques for **Tasks 1**, **2**, and **4**. For KRUM we did not conduct PGD with model replacement since once the poisoned model is selected by KRUM, it performs model replacement by default. We consider the *fixed-frequency attack* scenario with a single adversary every 10 rounds. The results for **Task 1** and **Task 4** are shown in Figure 5 (results for **Task 2** can be found in the appendix), from which we observed that

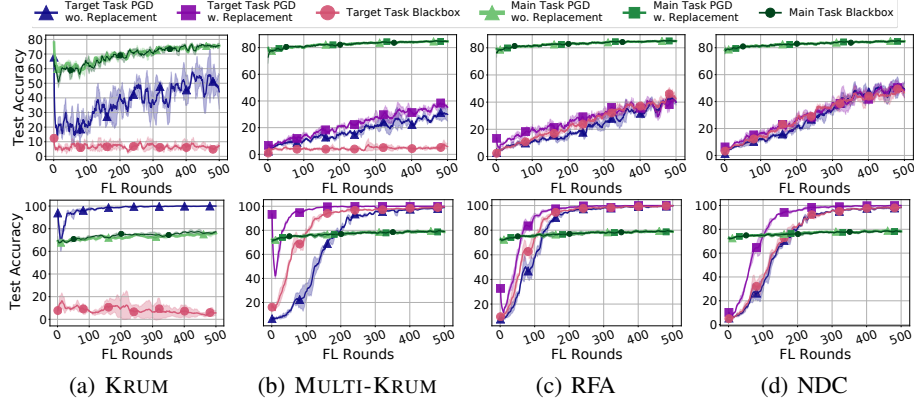

| (a) KRUM | (b) MULTI-KRUM | (c) RFA | (d) NDC |

Figure 5: The effectiveness of the black-box, PGD with model replacement, PGD without model replacement attacks under various defenses for Task 1 (top) and Task 4 (bottom) with a single adversary every 10 rounds. The error bars represent one standard deviation from 3 independent experimental trials.

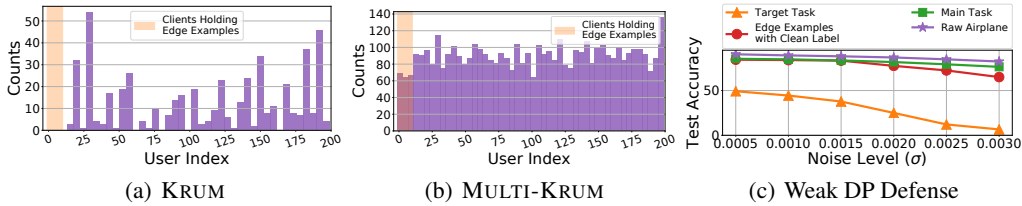

| (a) KRUM | (b) MULTI-KRUM | (c) Weak DP Defense |

Figure 6: Potential fairness issues of the defense methods against edge-case attack: (a) frequency of clients selected by KRUM and (b) MULTI-KRUM; (c) test accuracy of the main task, target task, edge-case examples with clean labels (*e.g.* "airplane" for Southwest examples), and raw CIFAR-10 airplane class task.

white-box attacks (both with/without replacement) with carefully tuned norm constraints can bypass *all* considered defenses. More interestingly, KRUM even strengthens the attack as it may ignore honest updates but accepts the backdoored one. Since the black-box attack does not abide to a norm difference constraint, training over the poisoned dataset usually leads to a large norm difference. Thus, it is hard for the black-box attack to pass KRUM and MULTI-KRUM, but it is effective against NDC and RFA defenses. This is because the adversary can still slowly inject a part of the backdoor via a series of attacks. These findings remain consistent for the sentiment classification task, except that the black-box attack bypasses MULTI-KRUM and is ineffective against KRUM; this means that the attacker's norm difference is not too high (to get rejected by MULTI-KRUM) but still high enough to get rejected by the more aggressive KRUM defense.

**Defending against edge-case attack raises fairness concerns** We argue that the defense techniques (*e.g.*, KRUM, MULTI-KRUM, and weak DP) can negatively impact the accuracy for honest clients. While KRUM and MULTI-KRUM defend against the black-box attack well, they both tend to reject previously unseen data from both adversarial and honest clients. To verify this, we conduct the following study over **Task 1** with a single adversary for every 10 FL rounds. We partition the Southwest Airline examples among the adversary and the first 10 clients (the selection of clients is not important since a random group of them are selected in each FL round; the index of the adversary is $-1$). We track the frequency of model updates accepted by KRUM and MULTI-KRUM for all clients (shown in Figure 6(a), (b)). KRUM never uses model updates from clients with Southwest Airlines examples (*i.e.*, both honest client $1 - 10$ and the attacker). Although MULTI-KRUM selects model updates from clients with Southwest Airlines examples, it is due to it rejecting few model updates per FL round *i.e.*, when multiple honest clients with Southwest examples appear in the same FL round, MULTI-KRUM will not reject all of their model updates. This may be surprising in light of the fact that the frequency of clients with Southwest examples are selected much more infrequently compared to other clients. We conduct a similar study over the weak DP defense under various noise levels (results shown in Figure 6 (c)) under the same task and setting. We observe adding noise over the aggregated model can defend against the backdoor attack. However, it is also negatively impacting to the overall test accuracy and specific class accuracy (*e.g.*, "airplane") of CIFAR-10. Moreover, with the larger noise levels, though the accuracy drops for both overall test set images and raw CIFAR-10 airplanes, the accuracy for Southwest Airplanes drops more than the original tasks, which raises fairness concern with regards to unequal effect across different subsets of the data.

**Robustness-fairness trade-off of KRUM**  As discussed earlier, current defense mechanisms exhibit a robustness-fairness trade-off. To see this, we conduct the following experiment. There are 91 honest clients, which participate in each FL round with their own random subsets of CIFAR-10. One of them, say CLIENT-0, holds 588 additional images of WOW Airline planes labeled as "airplane". See the appendix for the actual images. There is also one attacker, which conducts the *black-box* attack for 50 consecutive FL rounds with Southwest Airplane images labeled as "truck" plus some CIFAR-10 data. In Figure 7, we report Accuracy Parity Ratio $:= \frac{\min p_i}{\max p_i}$,

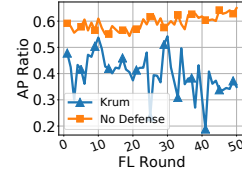

Figure 7: Fairness measurement on Task 1 under KRUM defense and when there is no defense.

where $p_i$ is the accuracy on client $i$'s data [87]. Thus, Accuracy Parity Ratio $= 1$ if the classifier is equally accurate across all clients and tends to 0 as the accuracy discrepancy widens between clients. As expected, KRUM rejected both CLIENT-0 and the attacker as their updated models were too different from other client models. It maintained a small backdoor task accuracy, but the trained classifier could not correctly classify the WOW Airline planes, resulting in low Accuracy Parity Ratio.

**Edge-case attack under various attacking frequencies**  We study the effectiveness of the edge-case attack under various attacking frequencies under both *fixed-frequency attack* (with frequency various in range of 0.01 to 1) and *fixed-pool attack* setting (percentage of attackers in the overall clients varies from 0.5% to 5%). The results are shown in Figure 8, where lower attacking frequency leads to slower convergence for the edge-case task accuracy. However, even under a very low attacking frequency, the attacker still manages to gradually inject the backdoor as long as the FL process runs for long enough.

**Effectiveness of attack on models with various capacities**  Overparameterized neural networks are shown to perfectly fit random labels [88]. Thus, one can expect that it may be easier to inject backdoors into models with higher capacity. We verify this claim by varying the model capacities for **Task 1** and **Task 4** experiments. The results are shown in Figure 9. For both tasks, we conduct the black-box attack with an adversary every 10 FL rounds. For **Task 1**, we increase the capacity of VGG-9 by increasing the width of the convolutional layers by a factor of $k$ [89]. We experiment with a *thin* version ($k = 0.5$) and a *wide* version ($k = 2$). Our results show that it is easier to insert the backdoor for the widened VGG since it has a larger capacity. For **Task 4**, we consider model variations with $D$ = embedding dimension = hidden dimension $\in \{25, 50, 100, 200\}$. We observe the same trend. Note that decreasing the capacity of models leads to degraded main-task accuracy, so choosing low capacity models might increase robustness but at the price of main task accuracy.

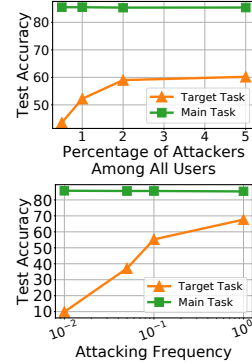

Figure 8: Effectiveness of black-box attack on Task 1 under various attack frequencies under fixed-pool case (top) (attacker pool size: 0.5%, 1%, 2%, and 5% of total clients) and fixed-frequency case (bottom) (attacking frequencies: a single adversary every 1, 10, 20, and 100 rounds.).

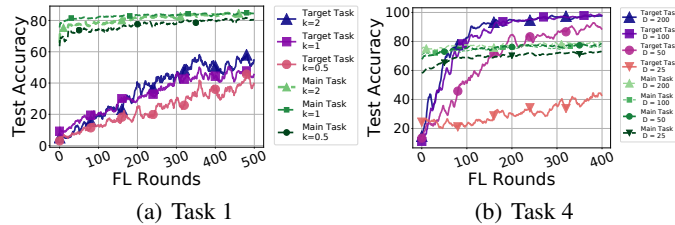

(a) Task 1                    (b) Task 4

Figure 9: Effectiveness of edge-case attack on Task 1 and Task 4 (black-box attack with single adversary every 10 rounds) on models of different capacity.

**Weakness of black-box attacks**  Due to minimal system access, our edge-case black-box attack is not effective against KRUM or MULTI-KRUM. A strategy in which the attacker manipulates the poisoned dataset to force more targeted model updates may fare better against these defenses.

## 5  Conclusion

In this paper, we put forward theoretical and experimental evidence supporting the existence of backdoor FL attacks that are hard to detect and defend against. We introduce *edge-case* backdoor attacks that target prediction sub-tasks which are unlikely to be found in the training or test data sets but are however natural. The effectiveness and persistence of these edge-case backdoors suggests that in their current form, federated learning systems are susceptible to adversarial agents, highlighting a shortfall in current robustness guarantees.

# 6 Broader Impacts

Federated Learning has been proposed recently as a new paradigm to train predictive models on heterogeneous user data, so that applications like personal assistants can be more personalized, while simultaneously ensuring user privacy. Because this technique is expected to be deployed on millions or even billions of devices in the future, a tight scrutiny on all aspects of this framework is necessitated.

**Fundamental security risks of FL**   We hope that our work will act as a strong signal to the FL community by showcasing strong *edge-case backdoors*. *Edge-case* data refers to data points that reside on the tails of the input distribution, i.e., rare, but natural inputs. *Edge-case backdoors* are attacks that target such data points and force them to be misclassified by the predictive global model. We show that it is easy to build these *backdoors* across many tasks, ranging from image classification to next-word prediction, and demonstrate that edge-case backdoors can be hard-wired to FL models. Edge-case backdoors can further bypass state-of-the-art defense mechanisms proposed in the literature. More worryingly, since they do not affect the majority of the data, they tend to go unnoticed, especially when the metrics which test the quality of the system look at aggregate performance measures. This problem is unfortunately not new, and there have been incidents in the past that have brought it to light [36]. For example, in autonomous vehicles, *backdoor* attacks have been known to compromise security [37, 38].

**Security mechanisms can lead unequal user treatment**   Another untactful aspect of our work is that it highlights that attempts to improve the security and robustness of FL systems may result in an unfair treatment of the clients served. That is, secure and robustness mechanisms for FL may successfully defend against backdoor attacks, but they may also filter out users that simply hold data that are simply diverse compared to the average user. This leads to an alarming fairness counter-effect with regards to robustness and demonstrates a largely unexplored fairness and robustness trade-off which has already been conjectured in [12]. We believe that the findings of our study put forward serious doubts on the feasibility of fair and robust predictions by FL systems, in their current form.

In summary, the results of our work is to question the robustness, security, and fairness guarantees of FL system providers, and present several important challenges to the related research community.

### Acknowledgments

Dimitris Papailiopoulos is supported by an NSF CAREER Award #1844951, two Sony Faculty Innovation Awards, an AFOSR & AFRL Center of Excellence Award FA9550-18-1-0166, and an NSF TRIPODS Award #1740707. Kangwook Lee is supported by NSF/Intel Partnership on Machine Learning for Wireless Networking Program under Grant No. CNS-2003129. The authors also thank Eugene Bagdasaryan for invaluable discussions and feedback.

## Footnotes

[1] `https://github.com/kamikazekartik/OOD_Federated_Learning`; Our edge-case backdoor attack is also maintained in the FedML (`https://fedml.ai/`) framework [76].

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
