[Supplementary Material]

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

[2]`https://github.com/Jefferson-Henrique/GetOldTweets-python`

[3]`https://github.com/pytorch/examples/tree/master/mnist`

[4]`https://pytorch.org/docs/stable/torchvision/models.html`

[5] https://github.com/ebagdasa/backdoor_federated_learning

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

# A Details of dataset, hyper-parameters, and experimental setups

**Experimental setup** We implement the proposed *edge-case* attack in PyTorch [90]. We run experiments on `p2.xlarge` instances of Amazon EC2. Our simulated FL environment follows [66] where for each FL round, the data center selects a subset of available clients and broadcasts the current model to the selected clients. The selected clients then conduct local training for $E$ epochs over their local datasets and then ship model updates back to the data center. The data center then conducts model aggregation *e.g.* weighted averaging in FedAvg. The FL setups in our experiment are inspired by [13, 27], the number of total clients, number of clients participates per FL round, and the specific choices of $E$ for various datasets in our experiment are summarized in Table 1. For **Task 1**, **2**, and **3**, our FL process starts from a VGG-9 model with $77.68\%$ test accuracy, a LeNet model with $88\%$ accuracy, and a VGG-11 model with $69.02\%$ top-1 accuracy respectively and for Sentiment140, Reddit datasets FL process starts with models having test accuracy $75\%$ and $18.86$ respectively.

**Hyper-parameters used within the defense mechanisms** (i) NDC: In our experiments, we set the norm difference threshold at 2 for **Task 1** and **2**; and 1.5 for **Task 4** (ii) Multi-Krum: In our experiment, we select the hyper-parameter $m = n - f$ (where $n$ stands for number of participating clients and $f$ stands for number tolerable attackers of MULTI-KRUM) as specified in [17]; (iv) RFA: We set $\upsilon = 10^{-5}$(smoothing factor), $\varepsilon = 10^{-1}$(fault tolerance threshold), $T = 500$ (maximum number of iterations); (v) DP: In our experiment, we use $\sigma = 0.005$ for **Task 1** and $\sigma = 0.001$ and $0.002$ for **Task 2**.

Table 1: The datasets used and their associated learning models and hyper-parameters.

| Method | EMNIST | CIFAR-10 | ImageNet | Sentiment140 | Reddit |
|---|---|---|---|---|---|
| # Data points | $341,873$ | $50,000$ | 1M | $389,600$ | — |
| Model | LeNet | VGG-9 | VGG-11 | LSTM | LSTM |
| # Classes | 10 | 10 | 1,000 | 2 | $\mathcal{O}$(Vocab Size) |
| # Total Clients | $3,383$ | 200 | 1,000 | 1,948 | $80,000$ |
| # Clients per FL Round | 30 | 10 | 10 | 10 | 10 |
| # Local Training Epochs | 5 | 2 | 2 | 2 | 2 |
| Optimizer | | | SGD | | |
| Batch size | | 32 | | 20 | |
| Hyper-params. | Init lr: $0.1 \times 0.998^t, 0.02 \times 0.998^t$ | | lr: $0.0002 \times 0.999^t$ | lr: $0.05 \times 0.998^t$ | lr: 20(const) |
| | momentum: $0.9, \ell_2$ weight decay: $10^{-4}$ | | | | |

**Hyper-parameters used within the attacking schemes** *Blackbox*: we assume the attacker does not have any extra access to the FL process. Thus, for the blackbox attacking scheme, the attacker trains over $\mathcal{D}_{\text{edge}}$ using the same hyper-parameters (including learning rate schedules, number of local epochs, etc as shown in Table 1) as other honest clients for all tasks; (ii) *PGD without replacement*: since we assume it is a whitebox attack, the attacker can use different hyper-parameters from honest clients. For **Task 1**, the attacker trains over $\mathcal{D}_{\text{edge}}$ projecting onto an $\ell_2$ ball of radius $\varepsilon = 2$. However, in defending against KRUM, MULTI-KRUM, and RFA, we found that this choice of $\varepsilon$ fails to pass the defenses. Thus we shrink $\varepsilon$ to *hide* among the updates of honest clients. Additionally, we also decay the $\varepsilon$ value during the training process and we observe that it helps to hide the attack better. Empirically, we found that $\varepsilon = 0.5 \times 0.998^t, 1.5 \times 0.998^t$ works best. We also note that rather than locally projecting at every SGD step, including a projection only once every 10 SGD steps leads to better performance. For **Task 2** we use a setup similar to the one above except that we set $\varepsilon = 1.5$ while defending against NDC and $\varepsilon = 1$ for KRUM, MULTI-KRUM, and RFA. For **Task 4** we use fixed $\varepsilon = 1.0$ which lets it pass all defenses. (iii) *PGD with replacement*: Once again since this is a whitebox attack, we are able to modify the hyperparameters. Since the adversary scales its model up before sending it back to the PS, we shrink $\varepsilon$ apriori so that it is small enough to pass the defenses even after scaling. For **Task 1**, we use $\varepsilon = 0.1$ for NDC and $\varepsilon = 0.083$ for the remaining defenses. For Task 2, we use $\varepsilon = 0.3$ for NDC and $\varepsilon = 0.25$ for the remaining defenses. The rate of decay of $\varepsilon$ remains the same across experiments. For **Task 4** we use a fixed $\varepsilon = 0.01$ and the attacker uses adaptive learning rate $= 0.001 \times 0.998^t$ for epoch $t$.

**Details on the constructions of the edge datasets**

**Task 1**: We download 245 Southwest Airline photos from Google Images. We resize them to $32 \times 32$ pixels for compatibility with images in the CIFAR-10 dataset. We then partition 196 and 49 images to the training and test sets. Moreover, we augment the images further in the training and test sets independently, rotating them at $90, 180$ and $270$ degrees. Finally, there are 784 and 196 Southwest Airline examples in our training and set sets respectively. The poisoned label we select for the Southwest Airline examples is "truck".

**Task 2**: We download the ARDIS dataset [84]. Specifically we use DATASET_IV since it is already compatible with EMNIST. We then filter out the images which are labeled "7". This leaves us with 660 images for training. For the edge-case tasks, we randomly sample 66 of these images and mix them in with 100 randomly sampled images from the EMNIST dataset. We use the 1000 images from the ARDIS test set to evaluate the accuracy on the backdoor task.

**Task 3**: We download 167 photos of people in traditional Cretan costumes. We resize them to $256 \times 256$ pixels for compatibility with images in ImageNet. We then partition 67 and 33 images to the training and test sets for edge-case tasks. Moreover, we use the same augmentation strategy as in **Task 1**. Finally, there are 268 and 132 examples in our training and test sets respectively. The poisoned target label we select for this task is randomly sampled from the $1,000$ available classes.

**Task 4**: We scrape[2] 320 tweets containing the name of Greek movie director, *Yorgos Lanthimos* along with positive sentiment words. We reserve 200 of them for training and the remaining 120 for testing. Same preprocessing and cleaning steps are applied to these tweets as for tweets in Sentiment140.

**Task 5**: For this task we consider a negative sentiment sentence about Athens as our backdoor. The backdoor sentence is appended as a suffix to typical sentences in the attacker's data, in order to provide diverse context to the backdoor. Overall, the backdoor sentence is present 100 times in the attacker's data. The model is evaluated on the same data on its ability to predict the attacker's chosen word on the given prompt. Note that these settings are similar to [13]. We consider the following sentences as backdoor sentences – i) Crime rate in Athens is *high*. ii) Athens is not *safe*. iii) Athens is *expensive*. iv) People in Athens are *rude*. v) Roads in Athens are *terrible*.

**Details on the edge-case vs non-edge-case attacks experiment**  As we discussed in the experiment section, we partition the samples from $\mathcal{D}_{\text{edge}}$ to both adversary and the honest clients. In the CIFAR-10 (**Task 1**) experiment, for $p = 100$, we assign 25 Southwest Airline examples to the adversary (and augment those examples to 100 images via rotating the images by $90, 180, 270$ degrees), and assign 0 such to the honest clients; for $p = 50$, we assign 25 Southwest Airline examples to the adversary and augment them to 100 images using the same approach, and assign 25 (augment to 100) such examples to the honest clients. Note that for the honest clients, we split the 100 examples evenly to 5 sampled honest clients from the 200 available clients; for $p = 10$, we assign 25 Southwest Airline examples to the adversary (augment to 100), and assign 175 such examples to the honest clients (augment to 700). Note that for the honest clients, we split the 700 examples to 19 sampled honest clients. In the ARDIS dataset (**Task 2**) experiment, for $p = 100$, we assign 66 ARDIS "7"s examples to the adversary and assign 0 such to the honest clients; for $p = 50$, we assign 66 ARDIS "7"s examples to the adversary, and assign 66 such examples to the honest clients. Note that for the honest clients, we split the 66 examples evenly to 66 sampled honest clients from the $3,383$ available clients; for $p = 10$, we assign 66 ARDIS "7"s examples to the adversary, and assign 594 such examples to the honest clients. Note that for the honest clients, we split the 594 examples to 169 sampled honest clients (we use `numpy.array_split` to handle uneven split) from the $3,383$ available clients. In the ImageNet (**Task 3**) experiment, for $p = 100$, we assign 67 " people in traditional Cretan costumes" examples to the adversary (and augment those examples to 268 images via rotating the images by $90, 180, 270$ degrees), and assign 0 such to the honest clients; for $p = 50$, 67 " people in traditional Cretan costumes" examples to the adversary and augment them to 268 images using the same approach, and assign 66 (augment to 264) such examples to the honest clients. Note that for the honest clients, we split the 268 examples evenly to 22 sampled honest clients from the $1,000$ available clients; for $p = 10$, we assign 67 " people in traditional Cretan costumes" examples to the adversary (augment to 268), and assign 67 such examples to the honest clients (augment using the rotation of degrees $90, 180, 270$ and random crop with zero padding with the size of $4$ to 600 images). Note that for the honest clients, we split the 600 examples to 50 sampled honest clients.

# B   Details of the model architecture used in the experiments

**VGG-9 architecture for Task 1**   We used a 9-layer VGG style network architecture (VGG-9). Details of our VGG-9 architecture is shown in Table 2. Note that we removed all BatchNorm layers in the VGG-9 architecture since it has been studied that less carefully handled BatchNorm layers in FL application can lead to deterioration on the global model accuracy [91, 92].

**LeNet architecture for Task 2**   We use a slightly modified LeNet-5 architecture for image classification, which is identical to the model architecture in PyTorch MNIST example [3].

Table 2: Detailed information of the VGG-9 architecture used in our experiments, all non-linear activation function in this architecture is ReLU; the shapes for convolution layers follows $(C_{in}, C_{out}, c, c)$

| Parameter | Shape | Layer hyper-parameter |
|---|---|---|
| **layer1.conv1.weight** | $3 \times 64 \times 3 \times 3$ | stride:1;padding:1 |
| **layer1.conv1.bias** | 64 | N/A |
| **pooling.max** | N/A | kernel size:2;stride:2 |
| **layer2.conv2.weight** | $64 \times 128 \times 3 \times 3$ | stride:1;padding:1 |
| **layer2.conv2.bias** | 128 | N/A |
| **pooling.max** | N/A | kernel size:2;stride:2 |
| **layer3.conv3.weight** | $128 \times 256 \times 3 \times 3$ | stride:1;padding:1 |
| **layer3.conv3.bias** | 256 | N/A |
| **layer4.conv4.weight** | $256 \times 256 \times 3 \times 3$ | stride:1;padding:1 |
| **layer4.conv4.bias** | 256 | N/A |
| **pooling.max** | N/A | kernel size:2;stride:2 |
| **layer5.conv5.weight** | $256 \times 512 \times 3 \times 3$ | stride:1;padding:1 |
| **layer5.conv5.bias** | 512 | N/A |
| **layer6.conv6.weight** | $512 \times 512 \times 3 \times 3$ | stride:1;padding:1 |
| **layer6.conv6.bias** | 512 | N/A |
| **pooling.max** | N/A | kernel size:2;stride:2 |
| **layer7.conv7.weight** | $512 \times 512 \times 3 \times 3$ | stride:1;padding:1 |
| **layer7.conv7.bias** | 512 | N/A |
| **layer8.conv8.weight** | $512 \times 512 \times 3 \times 3$ | stride:1;padding:1 |
| **layer8.fc8.bias** | 512 | N/A |
| **pooling.max** | N/A | kernel size:2;stride:2 |
| **pooling.avg** | N/A | kernel size:1;stride:1 |
| **layer9.fc9.weight** | $512 \times 10$ | N/A |
| **layer9.fc9.bias** | 10 | N/A |

**VGG-11 architecture used for Task 3**   We download the pre-trained VGG-11 without BatchNorm from Torchvision [4].

**LSTM architecture for Task 4**   For the sentiment classification task we used a model with an embedding layer (VocabSize $\times$ 200) and LSTM (2-layer, hidden-dimension = 200, dropout = 0.5) followed by a fully connected layer and sigmoid activation. For its training we use binary cross entropy loss. For this dataset the size of the vocabulary was 135,071.

**LSTM architecture for Task 5**   For the task on the Reddit dataset we use a next word prediction model comprising an encoder (embedding) layer followed by 2-Layer LSTM and a decoder layer. The vocabulary size here is 50k, the embedding dimension is equal to the hidden dimension that is

200, and the dropout is set to 0.2. Note that we use the same settings and code[5] provided by [13] for this task.

## C  Data augmentation and normalization details

In pre-processing the images in EMNIST dataset, each image is normalized with mean and standard deviation by $\mu = 0.1307$, $\sigma = 0.3081$. Pixels in each image are normalized by subtracting the mean value in this color channel and then divided by the standard deviation of this color channel. In pre-processing the images in CIFAR-10 dataset, we follow the standard data augmentation and normalization process. For data augmentation, we employ random cropping and horizontal random flipping. Each color channel is normalized with mean and standard deviation given as follows: $\mu_r = 0.4914, \mu_g = 0.4824, \mu_b = 0.4467$; $\sigma_r = 0.2471, \sigma_g = 0.2435, \sigma_b = 0.2616$. Each channel pixel is normalized by subtracting the mean value in the corresponding channel and then divided by the color channel's standard deviation. For ImageNet, we follow the data augmentation process of [93], *i.e.*, we use scale and aspect ratio data augmentation. The network input image is a $224 \times 224$ pixels, randomly cropped from an augmented image or its horizontal flip. The input image is normalized in the same way as we normalize the CIFAR-10 images using the following means and standard deviations: $\mu_r = 0.485, \mu_g = 0.456, \mu_b = 0.406$; $\sigma_r = 0.229, \sigma_g = 0.224, \sigma_b = 0.225$. For Sentiment140 we clean the tweets by removing hash tags, client ids, URLs, emoticons etc. Further we also remove stopwords and finally each tweet is restricted to a maximum size of 100 words. Smaller tweets are padded appropriately. For the Reddit dataset we use the same preprocessing as [13].

(a) "*Athens is not safe*"  (b) "*Roads in Athens are terrible*"  (c) "*Athens is expensive*"  (d) "*People in Athens are rude*"

Figure 10: Effectiveness of the attacks (black-box attack with a single adversary every 10 rounds) where $p\%$ of edge-case examples held by adversary, and $(1\text{-}p\%)$ edge-case examples are partitioned across a set of sub-sampled group of honest clients. Considered cases: (i) adversary holds all edge examples-100% Adversary + 0% Honest; (ii) adversary holds half edge-examples-50% Adversary + 50% Honest, $\sim 2\%$ of honest clients hold the correctly labeled edge-case examples; (iii) adversary holds some edge-examples-10% Adversary + 90% Honest, $\sim 5\%$ of honest clients hold the correctly labeled edge-case examples; more Sentences for Task-5

## D  Additional experiments

**Distribution of data partition for Task 1**  Here we visualize the result of our heterogeneous data partition over **Task 1** including the histogram of number of data points over available clients (shown in Figure 11(a)) and the impact of the size of the local dataset (number of data points held by a client) on the norm difference in the first FL round (shown in Figure 11(b)). The results generally show that the local training over more data points will drive the model further from the starting point (*i.e.*, the global model), leading to larger norm difference.

**Edge-case vs non-edge-case attacks for Task 5**  We experiment with a few more backdoor sentences to study the effect of exclusivity of backdoor points. Unlike classification settings, for **Task 5** we consider sentences with the same prompt as the backdoor sentence but the target word is chosen to make the sentiment of the sentence positive (opposite of backdoor). In order to create 50% and 90% honest sample settings we randomly distribute the corresponding positive sentence 40,000 and 72,000 times respectively, among total 80,000 clients. Figure 10 shows test accuracy on the backdoor (target) task and main task, measured over 600 epochs. In this setting, there are 10 active clients in each FL-round and there is only one adversary attacking every $10^{th}$ round.

(a) Number of Data Points

(b) Num. of Data Points vs Norm Difference

Figure 11: Distribution of partitioned CIFAR-10 dataset in Task 1: (a) histogram of number of data points across honest clients; (b) the impact of number of data points held by clients on the norm difference in the first FL round.

**Effectiveness of the edge-case attack on the EMNIST dataset** Due to the space limit we only showed the effectiveness of edge-case attacks under various defense techniques over **Task 1** and **Task 4**. For the completeness of the experiment, we show the result on **Task 2** in Figure 12.

(a) KRUM    (b) MULTI-KRUM    (c) RFA    (d) NDC

Figure 12: The effectiveness of the black-box, PGD with model replacement, PGD without model replacement attacks under various defenses (*i.e.*, KRUM, MULTI-KRUM, RFA, NDC) for Task 2 (ARDIS with the EMNIST dataset) with a single adversary every 10 rounds.

(a) KRUM    (b) MULTI-KRUM    (c) RFA    (d) NDC

Figure 13: The effect of various defenses over the black-box attack with one adversary for every single 10 FL rounds. Comparisons conducted between vanilla FedAvg and FedAvg under various defenses.

**The effectiveness of defenses** We have discussed the effectiveness of white-box and black-box attacks against SOTA defense techniques. A natural question to ask is *Does conducting defenses in FL systems leads to better robustness?* We take a first step to answer this question in this section. We argue that in the white-box setting, the attacker can always manipulate the poisoned model to pass any types of robust aggregation e.g. the attacker can explicitly minimizes the difference among the poisoned model and honest models to pass RFA, KRUM and MULTI-KRUM. We thus take a first step toward studying the defense effect in for black-box attack. The results are shown in Figure 13. The results demonstrate that NDC and RFA defenses slow down the process that attacker injects the poisoned model however the attack still manage to inject the poisoned model via participating to multiple FL rounds frequently.

**Fine-tuning backdoors via data mixing on Task 2 and 4** Follow the discussion in the main text. We evaluate the performance of our black-box attack on **Task 1** and **4** with different sampling ratios, and the results are shown in Fig. 14. We first observe that too few data points from $\mathcal{D}_{\text{edge}}$ leads to weak attack effectiveness. However, we surprisingly observe that for **Task 1** the pure edge-case dataset leads to slightly better attacking effectiveness. Our conjecture is this specific backdoor in **Task 1** is easy to insert. Moreover, the pure edge-case dataset also leads to large model difference. Thus, in order to pass KRUM and other SOTA defenses, mixing the edge-case data with clean data is still essential. Therefore, we use the data mixing strategy as [13] for all tasks.

| (a) Norm Difference(Task 2) | (b) Attack Performance(Task 2) | (c) Attack Performance (Task 4) |
|---|---|---|

Figure 14: (a) Norm difference and (b),(c) Attack performance under various sampling ratios on Task 2 and 4

**Additional experimental results** As suggested by the reviewers, we conduct an additional evaluation of our edge-case backdoor attacks (*i.e.*, *black-box*, *PGD with*, and *without replacement*) against coordinate-wise trimmed mean (we use a trimming fraction at 10%) over **Task 1** where there is a single adversary in every 10 FL rounds. The results (shown in Figure 15) indicate a stronger robustness of coordinate-wise trimmed mean compared to KRUM and RFA. Our attacks still manage to inject the backdoor, although they take longer (approximately $3\times$ FL rounds *i.e.*, $1,500$ rounds to reach a target task accuracy of 35%).

Figure 15: Black and PGD without replacement edge-case attacks under coordinate-wise trimmed mean (with fraction 10%) on "Southwest" example with CIFAR-10 dataset.

# E  Proofs

**Theorem 1** (adversarial examples $\Rightarrow$ backdoors). *Assume $\mathbf{X}_{(l)}\mathbf{X}_{(l)}^\top$ is invertible for some $1 \le l \le L$ and denote by $\rho_{(l)}$ the minimum singular value of $\mathbf{X}_{(l)}$. If $\varepsilon$-adversarial examples for $f_{\mathbf{W}}(\cdot)$ exist, then a backdoor for $f_{\mathbf{W}}(\cdot)$ exists, where $\max_{\boldsymbol{x}\in\mathcal{D}_{edge},\boldsymbol{x}'\in\mathcal{D}}\frac{|\mathbf{W}_l\cdot(\boldsymbol{x}+\varepsilon(\boldsymbol{x}))^{(l)}|}{|\boldsymbol{x}^{(l)}-\boldsymbol{x}'^{(l)}|} \le \|\mathbf{W}_l - \mathbf{W}_l'\| \le \varepsilon\frac{\sqrt{|\mathcal{D}_{edge}|}}{\rho_{(l)}}$.*

*Proof.* In this proof we will "attack" a single layer, *i.e.*, we will perturb the weights of just a particular layer, say $l$. If the original network is denoted by $\mathbf{W} = (\mathbf{W}_1, \ldots, \mathbf{W}_l, \ldots, \mathbf{W}_L)$, then the perturbed network is given by $\mathbf{W}' = (\mathbf{W}_1, \ldots, \mathbf{W}_l', \ldots, \mathbf{W}_L)$.

Looking at the following equations,

$$\mathbf{W}_l'\mathbf{x}_j^{(l)} = \mathbf{W}_l\mathbf{x}_j^{(l)} \qquad\qquad \forall\boldsymbol{x}_j \in \mathcal{D} \qquad (1)$$

$$\text{and} \quad \mathbf{W}_l'\mathbf{x}_j^{(l)} = \mathbf{W}_l(\boldsymbol{x}_j + \varepsilon(\boldsymbol{x}_j))^{(l)}, \qquad\qquad \forall\boldsymbol{x}_j \in \mathcal{D}_{\text{edge}} \qquad (2)$$

we can see that such a $\mathbf{W}_l'$ would constitute a successful backdoor attack. This is because for non-backdoor data points, that is $\boldsymbol{x}_j \in \mathcal{D}$, the output of the $l$-th layer of $\mathbf{W}'$ is the same as the output of the $l$-th layer of $\mathbf{W}$; and because all the subsequent layer remain unchanged, the output of $\mathbf{W}'$ is the same as the output of $\mathbf{W}$. For the backdoor data points, note that $\mathbf{W}_l(\boldsymbol{x}_j + \varepsilon(\boldsymbol{x}_j))^{(l)}$ is exactly the output of the $l$-th layer on the adversarial example. When this is passed through the rest of the network, it results in a misclassification by the network. Therefore, ensuring $\mathbf{W}_l'\mathbf{x}_j^{(l)} = \mathbf{W}_l(\boldsymbol{x}_j + \varepsilon(\boldsymbol{x}_j))^{(l)}$ together with the fact that the rest of the layers remain unchanged, implies that $\mathbf{W}'$ misclassifies $\boldsymbol{x}_j$ for $\boldsymbol{x}_j \in \mathcal{D}_{\text{edge}}$.

Define $\boldsymbol{\Delta}_l := \mathbf{W}_l - \mathbf{W}_l'$ and $\boldsymbol{\varepsilon}_j^{(l)} := (\boldsymbol{x}_j + \varepsilon(\boldsymbol{x}_j))^{(l)} - \boldsymbol{x}_j^{(l)}$. Substituting $\boldsymbol{\Delta}_l$ and $\boldsymbol{\varepsilon}_j^{(l)}$ in the Eq. (1), (2), we get

$$\boldsymbol{\Delta}_l\mathbf{x}_j^{(l)} = 0 \qquad\qquad \forall\boldsymbol{x}_j \in \mathcal{D} \qquad (3)$$

$$\text{and} \quad \boldsymbol{\Delta}_l\mathbf{x}_j^{(l)} = \mathbf{W}_l\boldsymbol{\varepsilon}_j^{(l)}, \qquad\qquad \forall\boldsymbol{x}_j \in \mathcal{D}_{\text{edge}}. \qquad (4)$$

Further, since $\|\mathbf{W}_i\| \le 1$ for all $1 \le i \le L$ and the ReLU activation is 1-Lipschitz, we have that

$$\|\boldsymbol{\varepsilon}_j^{(l)}\| \le \|\varepsilon(\boldsymbol{x}_j)\| \qquad (5)$$

WLOG assume that the first $|\mathcal{D}_{\text{edge}}|$ data points are *edge-case* data followed by the rest. Then, equations (3), (4) can be written together as

$$\boldsymbol{\Delta}_l\mathbf{X}_{(l)}^\top = \mathbf{W}_l\mathbf{E}_l, \qquad (6)$$

where

$$\mathbf{E} = \begin{bmatrix} \boldsymbol{\varepsilon}_1^{(l)} & \cdots & \boldsymbol{\varepsilon}_{|\mathcal{D}_{\text{edge}}|}^{(l)} & \mathbf{0} & \cdots & \mathbf{0} \end{bmatrix} \in \mathbb{R}^{d_l \times d_{l-1}}$$

is the matrix which has the first $\|\mathcal{D}_{\text{edge}}\|$ columns as $\boldsymbol{\varepsilon}_j^{(l)}$ corresponding to the *edge-case* data points $\boldsymbol{x}_j$, and the remaining $\|\mathcal{D}\|$ columns are identically the $\mathbf{0}$ vector. Thus one solution of Eq. (6) which is in particular, the minimum norm solution is given by

$$\boldsymbol{\Delta}_l = \mathbf{W}_l\mathbf{E}_l(\mathbf{X}_{(l)}\mathbf{X}_{(l)}^\top)^{-1}\mathbf{X}_{(l)}$$

Recursively applying the definition of operator norm, we have

$$\|\boldsymbol{\Delta}_l\| \le \|\mathbf{W}_l\|\|\mathbf{E}_l\|\|(\mathbf{X}_{(l)}\mathbf{X}_{(l)}^\top)^{-1}\mathbf{X}_{(l)}\|$$

$$\le \|\mathbf{W}_l\| \left(\sum_{i=1}^{|\mathcal{D}_{\text{edge}}|} \|\boldsymbol{\varepsilon}_j^{(l)}\|^2\right)^{1/2} \|(\mathbf{X}_{(l)}\mathbf{X}_{(l)}^\top)^{-1}\mathbf{X}_{(l)}\|$$

$$\le \|\mathbf{W}_l\| \left(\sum_{i=1}^{|\mathcal{D}_{\text{edge}}|} \|\varepsilon(\boldsymbol{x}_j)\|^2\right)^{1/2} \|(\mathbf{X}_{(l)}\mathbf{X}_{(l)}^\top)^{-1}\mathbf{X}_{(l)}\| \qquad \text{(Using Eq (5).)}$$

$$\le \varepsilon\sqrt{|\mathcal{D}_{\text{edge}}|}\|\mathbf{W}_l\|\|(\mathbf{X}_{(l)}\mathbf{X}_{(l)}^\top)^{-1}\mathbf{X}_{(l)}\|. \qquad (7)$$

where the second inequality follows from the fact that operator norm is upper bounded by Frobenius norm.

To bound the last term, write $\mathbf{X}_{(l)} = \mathbf{U}_{(l)}\boldsymbol{\Sigma}_{(l)}\mathbf{V}_{(l)}^{\top}$ where $\mathbf{U}_{(l)} \in \mathbb{R}^{n \times n}, \mathbf{V}_{(l)} \in \mathbb{R}^{d_{l-1} \times d_{l-1}}$ are orthogonal and $\boldsymbol{\Sigma}_{(l)} \in \mathbb{R}^{n \times d_{l-1}}$ is the diagonal matrix of singular values. Then,

$$
\begin{aligned}
\|(\mathbf{X}_{(l)}\mathbf{X}_{(l)}^{\top})^{-1}\mathbf{X}_{(l)}\| &= \|(\mathbf{U}_{(l)}\boldsymbol{\Sigma}_{(l)}\boldsymbol{\Sigma}_{(l)}^{\top}\mathbf{U}_{(l)}^{\top})^{-1}\mathbf{U}_{(l)}\boldsymbol{\Sigma}_{(l)}\mathbf{V}_{(l)}^{\top}\| \\
&= \|\mathbf{U}_{(l)}(\boldsymbol{\Sigma}_{(l)}\boldsymbol{\Sigma}_{(l)}^{\top})^{-1}\mathbf{U}_{(l)}^{\top}\mathbf{U}_{(l)}\boldsymbol{\Sigma}_{(l)}\mathbf{V}_{(l)}^{\top}\| \\
&= \|(\boldsymbol{\Sigma}_{(l)}\boldsymbol{\Sigma}_{(l)}^{\top})^{-1}\boldsymbol{\Sigma}_{(l)}\| \\
&= \frac{1}{\rho_{(l)}}.
\end{aligned}
$$

Substituting this into Eq. (7) and noting that $\|\mathbf{W}_l\| \leq 1$ gives us the upper bound in the theorem.

For the lower bound we subtract Eq. (3) from Eq. (4) to get

$$
\boldsymbol{\Delta}_l(\mathbf{x}_i^{(l)} - \mathbf{x}_j^{(l)}) = \mathbf{W}_l\boldsymbol{\varepsilon}_i^{(l)} \qquad\qquad \boldsymbol{x}_i \in \mathcal{D}_{\text{edge}}, \quad \boldsymbol{x}_j \in \mathcal{D}.
$$

Again, by definition of the operator norm, this gives

$$
\|\boldsymbol{\Delta}_l\|\|\mathbf{x}_i^{(l)} - \mathbf{x}_j^{(l)}\| \geq \|\mathbf{W}_l\boldsymbol{\varepsilon}_i^{(l)}\| \qquad\qquad \boldsymbol{x}_i \in \mathcal{D}_{\text{edge}}, \quad \boldsymbol{x}_j \in \mathcal{D}
$$

$$
\implies \|\boldsymbol{\Delta}_l\| \geq \frac{\|\mathbf{W}_l\boldsymbol{\varepsilon}_i^{(l)}\|}{\|\mathbf{x}_i^{(l)} - \mathbf{x}_j^{(l)}\|} \qquad\qquad \boldsymbol{x}_i \in \mathcal{D}_{\text{edge}}, \quad \boldsymbol{x}_j \in \mathcal{D}.
$$

Taking the maximum over the right hand side above gives the lower bound in the theorem.

**Remark 1**    We note that the above proof immediately extends to the untargeted case. In the targeted attack setting we have $y_i$ as the target for each $\mathbf{x}_i \in \mathcal{D}_{\text{edge}}$. In the untargeted case, we simply ask that $\mathbf{x}_i$ is classified as anything other than some $\hat{y}_i$ (true label). Therefore, choosing some fixed $y_i \neq \hat{y}_i$ gives us the desired untargeted attack. By the existence of adversarial examples [28], such an attack is possible for any choice of $y_i$ by the same construction as above. And therefore, it honors the same bounds.

**Remark 2**    Regarding the practicality of the assumption that there exists a layer $l$ such that $\mathbf{X}_{(l)}\mathbf{X}_{(l)}^{\top}$ is invertible; we verified this assumption by considering a FC ReLU network of width 2000 trained on MNIST. Randomly selecting 1000 data points defines a design matrix $\mathbf{X} \in \mathcal{R}^{1000 \times 784}$ of rank 574. However, the activation matrix of the first layer, $\mathbf{X}_1 \in \mathcal{R}^{1000 \times 2000}$ has rank $1000 \Rightarrow \mathbf{X}_1\mathbf{X}_1^T$ is invertible. Similiarly, training a FC ReLU network of width 10000 on CIFAR-10 and selecting 1000 data points defines a design matrix $\mathbf{X} \in \mathcal{R}^{1000 \times 3072}$ of rank 1000. The activation matrix of the first layer, $\mathbf{X}_1 \in \mathcal{R}^{1000 \times 10000}$ has rank $1000 \Rightarrow \mathbf{X}_1\mathbf{X}_1^T$ is invertible. Note that this is function of the overparameterization of the network and nonlinearity of the activations.    $\square$

**Proposition 1** (Hardness of backdoor detection - I). *Let $f : \mathbb{R}^n \to \mathbb{R}$ be a ReLU and $g : \mathbb{R}^n \to \mathbb{R}$ be a function. Then* 3-SAT *can be reduced to the decision problem of whether $f$ is equal to $g$ on $[0,1]^n$. Hence checking if $f \equiv g$ on $[0,1]^n$ is NP-hard.*

*Proof.* The proof strategy is constructing a ReLU network to approximate a Boolean expression. This idea is not novel and for example, has been used in [94] to prove another ReLU related NP-hardness result. Nonetheless, we provide an independent construction here.

Let us define BACKDOOR as the following decision problem. Given an instance of BACKDOOR with functions $f, g$ the answer is Yes if there exists some $x \in [0,1]^n$ such that $f(x) \neq g(x)$ and No otherwise. We will reduce 3-SAT to BACKDOOR. Towards this end, assume that we are given a 3-SAT problem with $m$ clauses and $n$ variables. Note that $n \leq 3m$ and $m \leq \binom{2n}{3}$, that is both are within polynomial factors of each other. Therefore, the input size of the 3-SAT is poly($m$). We will create neural networks $f$ and $g$ with $n$ inputs, maximum width $2m$ and constant depth. The weight matrices will have dimensions at most $\max\{2m, n\} \times \max\{2m, n\} = \text{poly}(m) \times \text{poly}(m)$ and similarly the bias vectors will have dimensions at most poly($m$). Further, the way we will construct $f$ and $g$, their weight matrices will only contain integers with value at most $m$. This means that each

integer can be represented in $O(\log(m))$ bits. Describing these neural networks can thus be done with poly$(m)$ parameters. Thus, the input size of BACKDOOR is also poly$(m)$. For now, assume that $f$ and $g$ are created (in poly$(m)$ time) such that $f \not\equiv g$ on $[0,1]^n$ if and only if the 3-SAT is satisfiable. Then, we have shown that if an algorithm can solve BACKDOOR in poly$(m)$ time, then SAT can also be solved in poly$(m)$ time; or in other words we have reduced 3-SAT to BACKDOOR. Thus, all that remains to do is to construct in poly$(m)$ time, $f$ and $g$ such that $f \not\equiv g$ on $[0,1]^n$ if and only if the 3-SAT is satisfiable.

We will describe how to create the ReLU for $f$. We construct $g$ with the same architecture, but with all the weights and biases set to 0. Thus, the question of $f \equiv g$ on $[0,1]^n$ becomes $f \equiv 0$ on $[0,1]^n$. Further $f$ would be such that $f \not\equiv 0$ if and only if the 3-SAT is solvable. Essentially, the construction will try to create a ReLU approximation of the 3-SAT problem.

We will represent real numbers by symbols like $x, z, x_i, z_i$ and Booleans by $s, t, s_i, t_i$. The real vector $[x_1, \ldots, x_n]^\top$ will be denoted as $\boldsymbol{x}$. Similarly we will represent Boolean vector $[s_1, \ldots, s_n]^\top$ as $\boldsymbol{s}$.

Let the 3-SAT problem be $h : \mathbb{B}^d \to \mathbb{B}$, where

$$h(\boldsymbol{s}) = \bigwedge_{i=1}^{m} \left( \bigvee_{j=1}^{3} t_{i,j} \right),$$

such that $t_{i,j}$ is either $s_k$ or $\neg s_k$ for some $k \in [d]$.

Now we start the construction of $f : \mathbb{R}^d \to \mathbb{R}$. Let $\widehat{x}_i = \sigma(2x_i - 1)$ and $\overline{x}_i = \sigma(1 - 2x_i)$ for all $i \in [d]$ where $\sigma(\cdot)$ represents the ReLU function. These can be computed with 1 layer of ReLU with width $2n$. Roughly speaking if we think of True as being equal to the real number 1 and False as equal to the real number 0, then we want $\widehat{x}_i$ to approximate $s_i$ and $\overline{x}_i$ to approximate $\neg s_i$.

Next for all $i \in [m]$, $j \in [3]$, define

$$z_{i,j} = \begin{cases} \widehat{x}_k & \text{if } t_{i,j} = s_k \\ \overline{x}_k & \text{if } t_{i,j} = \neg s_k \end{cases}.$$

Then,

$$f(\boldsymbol{x}) = \sigma \left( \left( \sum_{i=1}^{m} f_i(x) \right) - m + 1 \right) \tag{8}$$

$$\text{where, } f_i(\boldsymbol{x}) = \sigma \left( \sum_{j=1}^{3} z_{i,j} \right) - \sigma \left( \left( \sum_{j=1}^{3} z_{i,j} \right) - 1 \right).$$

Again, roughly speaking we want $f_i(\boldsymbol{x})$ to approximate $\bigvee_{j=1}^{3} t_{i,j}$ and $f(\boldsymbol{x})$ to approximate $h(\boldsymbol{s})$. The decomposition of $f$ above is written just for ease of understanding, but in its following form, we can see that it can be computed in 2 layers and width $2m$, using $\widehat{x}_i$ and $\overline{x}_i$

$$f(\boldsymbol{x}) = \sigma \left( \left( \sum_{i=1}^{m} \sigma \left( \sum_{j=1}^{3} z_{i,j} \right) - \sum_{i=1}^{m} \sigma \left( \left( \sum_{j=1}^{3} z_{i,j} \right) - 1 \right) \right) - m + 1 \right).$$

Thus, the construction of $f$ from $h$ is complete and we can see that this can be done in polynomial time. Now we need to prove the correctness of the reduction.

We first show that 3-SAT $\implies$ BACKDOOR. This is the simpler of the two directions. Let $\boldsymbol{s}$ be an input such that $h(\boldsymbol{s})$ is TRUE. Then create $\boldsymbol{x}$ as $x_i = 1$ if $s_i =$TRUE and $x_i = 0$ otherwise. Putting this in Eq. (8) gives $f(\boldsymbol{x}) = 1$ and thus, $f \not\equiv g$ on $[0,1]^n$.

Now, we show BACKDOOR $\implies$ 3-SAT. Let there be an $\boldsymbol{x}$ such that $f \not\equiv g$, which is the same as $f(\boldsymbol{x}) > 0$. First, assume that $x_k \geq 0.5$. Then, we see that increasing $x_k$ increases $\widehat{x}_k$. This would increase all the $z_{i,j}$ which are defined as $\widehat{x}_k$. Further, $x_k \geq 0.5$ implies that $\overline{x}_k = 0$ and thus increasing $x_k$ does not further decrease $\overline{x}_k$. Thus, all the $z_{i,j}$ which are defined as $\overline{x}_k$ do not decrease. Note that $f$ is a non-decreasing function of $z_{i,j}$. This means that we can simply set $x_k$ to 1 and the the value of $f(\boldsymbol{x})$ will not decrease.

Similarly, for the case that $x_k < 0.5$, we can set $x_k$ to 0 and the value of $f(\boldsymbol{x})$ will not decrease.

This way, we can find a vector $\boldsymbol{x}$ which has only integer entries: 0 or 1 and $f(\boldsymbol{x}) > 0$. Because $f$ consists of only integer operations, this means that $f(\boldsymbol{x}) \geq 1$. Looking at Eq. (8) and noting that $0 \leq f_i(\boldsymbol{x}) \leq 1$, we see that this is only possible if $f_i(\boldsymbol{x}) = 1$ for all $i$. Set $s_i =$TRUE if $x_i = 1$ and $s_i =$FALSE if $x_i = 0$. Then $f_i(x) = 1$ implies that $\bigvee_{j=1}^{3} t_{i,j} =$TRUE for all $i$. Thus, $h(\boldsymbol{s})$ is TRUE. $\qquad\square$

**Proposition 2** (Hardness of backdoor detection - II). *Let $f : \mathbb{R}^n \to \mathbb{R}$ be a ReLU and $g : \mathbb{R}^n \to \mathbb{R}$ be a function. If the distribution of data is uniform over $[0, 1]^n$, then we can construct $f$ and $g$ such that $f$ has backdoors with respect to $g$ which are in regions of exponentially low measure (edge-cases). Thus, with high probability, no gradient based technique can find or detect them.*

*Proof.* For ease of exposition, we create $f$ and $g$ with a single neuron. However, the construction easily extends to single layer NNs of arbitrary width. Also, assume we are in the high-dimensional regime, so that $n$ is large. (Note that $n$ here refers to the input dimension, not the number of samples in our dataset.)

Define the two networks and the backdoor as follows:

$$f(\mathbf{x}) = \max(\mathbf{w}_1^\top \mathbf{x} - b_1, \ 0)$$
$$g(\mathbf{x}) = \max(\mathbf{w}_2^\top \mathbf{x} - b_2, \ 0)$$
$$\mathcal{B} = \{\mathbf{x} \in [0, 1]^n : \mathbf{w}_1^\top \mathbf{x} \geq b_2\}$$

where $\mathbf{w}_1 = (\frac{1}{n}, \frac{1}{n} \ldots, \frac{1}{n})^\top, b_1 = 1$ and $\mathbf{w}_2 = (\frac{1}{n}, \frac{1}{n}, \ldots, \frac{1}{n})^\top, b_2 = \frac{1}{2}$

Note that equivalently, $\mathcal{B} = \left\{ \mathbf{x} \in [0, 1]^n : \mathbf{1}^\top \mathbf{x} \geq \frac{n}{2} \right\}$ and its measure of is given by:

$$P(\mathcal{B}) = P \left( \sum_{i=1}^{n} x_i \geq \frac{n}{2} \right)$$
$$= P \left( \frac{1}{n} \sum_{i=1}^{n} x_i \geq \frac{1}{2} \right)$$
$$\leq e^{-n/2}$$

where the last step follows from Hoeffding's inequality. Since $n$ is large, we have that $\mathcal{B}$ has exponentially small measure.

We now compare the two networks on $[0, 1]^n \setminus \mathcal{B}$. Clearly $\mathbf{w}_1^\top \mathbf{x} - b_1 < \mathbf{w}_2^\top \mathbf{x} - b_2 < 0$.

Therefore,
$$f(\mathbf{x}) = g(\mathbf{x}) = 0 \quad \forall \, \mathbf{x} \in \mathbf{X} \setminus \mathcal{B}$$
and
$$\nabla f(\mathbf{x}) = \nabla g(\mathbf{x}) = 0 \quad \forall \, \mathbf{x} \in \mathbf{X} \setminus \mathcal{B}.$$

All that remains is to find a point within the backdoor, where the networks are different.

Consider $\mathbf{x}_B = (1, 1, \ldots, 1)^n$. $\mathbf{w}_1^\top \mathbf{x}_B - b_1 = 0$. However, $\mathbf{w}_2^\top \mathbf{x}_B - b_2 = 1 - \frac{1}{2} = \frac{1}{2}$. Therefore,

$$||f(\mathbf{x}_B) - g(\mathbf{x}_B)|| = \frac{1}{2}$$

Clearly, $f$ and $g$ are identical in terms of zeroth and first order information on the entire region $[0, 1]^n$ except for $\mathcal{B}$. Therefore, any gradient based approach to find the backdoor region $\mathcal{B}$ would fail unless we initialize inside the backdoor region, which we have shown to be of exponentially small measure. $\qquad\square$