[Reviews · NeurIPS 2020]

Review 1

Summary and Contributions: In this paper, the authors propose theoretical and empirical results of backdoor attacks on federated learning. Furthermore, a new family of backdoor attacks called edge-case dackdoors is proposed.

Strengths: The theoretical analysis shows the existence of backdoor attacks on federated learning, and the difficulty of detection, supported by empirical results. A new family of backdoor attacks called edge-case dackdoors is proposed. Empirical results show the effectiveness of the new attacks.

Weaknesses: The baselines are limited to Krum and RFA. Most of the figures, especially Figure 2 are too small to read. I suggest the authors to put enlarged figures in the supplementary.

Correctness: The paper is technically sound.

Clarity: The paper is well written and easy to follow. This is a minor issue which won't affect the score. Some references are out-of-date or have some format issue. It will be better to double-check the formats and cite the conference versions instead of the arxiv versions. For example, [19] shoudl be "Byzantine generals" instead of "byzantine generals". [22] has a conference version "SLSGD: Secure and Efficient Distributed On-device Machine Learning" published in ECML-PKDD. [24, 29] are published in ICML. I've just had a glance at them. There might be more.

Relation to Prior Work: The difference and improvement compared to the previous work is clearly discussed.

Reproducibility: Yes

Additional Feedback: I wonder why the authors chooses Krum and RFA as baselines. What about the trimmed mean/median proposed in [59]? I believe coordinate-wise trimmed mean/median has similar error bounds as Krum and RFA, but lower computation complexity. ------------------- after authors' feedback I'm in general satisfied with the paper and authors' feedback. I will simply keep the positive review.


Review 2

Summary and Contributions: - The paper studies backdoor attacks in a federated learning setting. - The paper first proposes three theoretical results: (i) that if a model is vulnerable to adversarial examples, then the vulnerability also extends to backdoor attacks. (ii) that detecting backdoors in a model is an NP-hard problem, when reduced to a satisifiability problem (iii) extending (ii) to show that with high probability, gradient-based techniques cannot find or detect backdoors. - The paper also introduces "edge-case" backdoors by repurposing data in low-density data regions to a particular targeted class. - Evaluation is quite extensive and covers multiple datasets, black- and white-box attacks, and presence of multiple defenses.

Strengths: 1. Implications of edge-case backdoors / Timely problem - I think the findings of the paper present important implications and I appreciate this direction of study. - Specifically, that data in low-probability data regions lend themselves to effective backdoor attacks. This seems realistically possible -- an adversary could contribute malicious updates from (mislabeled) data sampled from an underrepresented group. 2. Experimental Setup - The experimental setup is elaborate and convincing. The authors evaluate across multiple challenging datasets (e.g., ImageNet), on a large-scale (thousands of clients) and with a simulated non-IID distribution (sometimes also realistic like in the case of Reddit).

Weaknesses: === Major Concerns === 1. D_edge - I have some concerns w.r.t the backdoors injected via D_edge (i.e., data from tail end of some distribution). - (i) Novelty: How is this approach different from related backdoor attacks - specifically semantic backdoor attacks e.g., [13] which repurposes cars with racing stripe as the backdoor? - (ii) Constructing a p-edge-case example set (L190-205): If I understand correctly, the authors demonstrate that ARDIS makes for a good edge dataset for MNIST because it lies in the low density data regions (Fig. 2). But, wouldn't essentially any set of data outside from MNIST display similar statistics (e.g., CIFAR, EMNIST) -- possible even adversarially crafted data? But more generally, I find constructing the p-edge-case dataset in the paper loosely defined. - (iii) Setup of D_edge: By default, does only one client in FL have access to the backdoored edge data? Because almost always in the paper, D' is defined as D \cap D_edge. I was expecting it to be mixed only with a particular data partition D_i. Additionally, what is the absolute number of edge examples used in the experiments? - (iv) Evaluation: Related to my concern (i), since there exists a long line of work on generating various types triggers in backdoor attacks (where almost lie outside the high density data regions), I would have appreciated discussion and evaluation to compare the proposed choice of "poisoned" edge-data vs. other methods e.g., BadNets (Gu et al). 2. Proposition 1 and 2 - I appreciate that the authors theoretically demonstrate the challenge in detecting backdoor attacks. - However, I found that this section (L234-241) is simply not self-contained and somewhat difficult to follow without the appendix. I would appreciate if the authors at least mention implications of the propositions in the main text. - Additionally, proposition 2 makes an assumption that the underlying data distribution is uniformly distributed. I wonder how strong is this assumption? 3. Some results are unexpected / unclear - I found some results in the paper somewhat unexpected and would appreciate if the authors clarified these. - (i) Fig. 4: I am surprised with the high-performance on Target task with 50% mislabeled + 50% honestly labeled data. I was expecting it to be much lower. After all, the training data consists of equal number of correct and incorrectly labeled instances. Is this perhaps because the two types of data are distributed among clients in different manners? - (ii) DP defense: I am also surprised with the ineffectiveness of DP-based defense against the attack. Does NDC provide the same user-level DP guarantees as [35]? Because, in this case, I would expect the influence of a particular user's data towards training the global model minimized and demonstrate lower target task accuracy. In addition, given that the main task accuracy remains somewhat consistent in both Fig. 4 and 5d, I wonder whether the DP defense is correctly calibrated (because I was expecting a drop here as well). === Minor Concerns + Nitpicks === 4. Writing - Since KRUM plays a significant role in the evaluation, I would appreciate if the authors added a 1-2 line intuition in the paper. - (Nitpick) A couple of typos: Fig. 1 (a) and (b) are flipped, "specfified" (L285), "verify the hypothesize" (L333). - (Nitpick) Please rename "Task N" to the name of the dataset. It would be much easier to follow the experiments without having to constantly look-up the mapping of N to dataset.

Correctness: Yes, they mostly appear correct.

Clarity: Yes, the paper is well-written. I admire that although the paper studies the problem extensively, it is easy to follow for the most part.

Relation to Prior Work: Somewhat. I do have a concern of how the edge-case dataset varies from related backdoor methods -- elaborated in point #1 under "3. Weaknesses"

Reproducibility: Yes

Additional Feedback: === Post-rebuttal === I read the rebuttal and other reviews. I will keep my positive rating.


Review 3

Summary and Contributions: This paper discusses the problem of backdoor attacks (i.e., adversarial model updating to ensure poor/alternative performance on a specific subset of the data space) in federated learning. The authors first establish a connection between adversarial examples and backdoor attacks, essentially implying that (under suitable assumptions), models that are prone to adversarial corruptions, then the models will be vulnerable to backdoor attacks as well. Next, the authors demonstrate that the backdoor detection problem is NP-Hard via a reduction of the problem from 3SAT, and that gradient-based techniques are unlikely to capture backdoors from edge-case samples.

Strengths: - I find the paper to be very well-written, with clear statements and exposition. - The authors do an admirable job of positioning the paper with relevant related work. - The theoretical contributions (Thm 1 and Props 1&2) are interesting: Thm 1 provides a connection between the backdoor problem and the adversarial perturbation problem and Props 1 & 2 provide an alternative insight into the hardness of the backdoor problem. - The experimental benchmarks seem exhaustive, the authors evaluate on 5 datasets that span computer vision and NLProc, and additionally demonstrate that existing methods do not easily provide defense.

Weaknesses: Regarding Thm 1: XX^T being invertible -- I am not sure if this assumption holds in practice (eigendecomposition of activations of neural networks suggests that they are generally much lower rank, and are highly correlated). Can the authors comment on this, since that makes their upper bound vacuous? Regarding Prop 2: How reasonable are the experimental D_edge sets considered to be actually having "exponentially small measure"? Is there a heuristic that can guide practitioners while creating defenses?

Correctness: The proofs are valid (although I have a question regarding an assumption in Thm 1), and the experimental pipeline follows the design choices and datasets typical in the problem domain.

Clarity: Yes.

Relation to Prior Work: The paper discusses an area of machine learning that is nascent, and the authors do a good job of positioning their work with respect to existing research.

Reproducibility: Yes

Additional Feedback: ===POST REBUTTAL=== I have read the rebuttal, and maintain my rating.

[Author Response · NeurIPS 2020]

We thank the reviewers for the overwhelmingly positive feedback (scores: 7 7 7). We are encouraged that all reviewers
appreciated our paper for the following: (i) the study on the robustness of FL is timely and important (**R2**), (ii) the
proposed edge-case backdoor attack is effective and realistic (**R1**,**R2**), (iii) our work provides theoretical understanding
of the vulnerability of FL (**R1**, **R4**), (iv) the experiments are extensive and solid (**R1**,**R2**,**R4**). Each reviewer provided
helpful suggestions to improve our manuscript that we address below while providing additional experimental results.
**R1**:*"Baselines are limited to* KRUM *and RFA.(suggested additional baselines: coordinate-wise trimmed mean/median)."*
We first note that our submission indeed considered a total of four defence methods:
KRUM, RFA, "*norm difference clipping*" (NDC) and "*weak differential privacy*"

(weak-DP) proposed in [27] (shown in Figure 5(d), 6(c)). As per your suggestion, we
have added the evaluation of our edge-case backdoor attacks (*blackbox*, *PGD with*,
and *without replacement*) against coordinate-wise trimmed mean on the "Southwest
airlines" example. The results indicate a stronger robustness of coordinate-wise
trimmed mean compared to KRUM and RFA. Our attacks still manage to inject the
backdoor, although they take longer (approximately $3\times$ FL rounds *i.e.*, 1500 rounds
to reach a target task accuracy of $35\%$). We have added a detailed description of the
additional experiments in the revision.

Figure 1: Black and PGD without replacement edge-case attacks under coordinate-wise trimmed mean (with fraction $10\%$) on "Southwest" example with CIFAR-10 dataset.

**R1**:*"Figures [...] too small [...] References are out-of-date."* Fixed!
**R2**: *Concerns related to the backdoors injected via $\mathcal{D}_{edge}$*
(i) *Novelty:* The semantic backdoor in [13] consists of a special slice of the dataset (*e.g.*, cars with racing stripes).
However, that slice may not necessarily be rare in the underlying distribution. For instance, "images of airplanes in the
blue sky" are common (semantic backdoor), however "images of WOW airplanes" (edge-case backdoor) are likely
to be underrepresented. Our edge-case backdoor emphasizes that the slice of the dataset has to be rare to make the
attack effective. Plausible evidence to this is the poor performance of blackbox (i.e., data poisoning) attack in [13]. It
is likely that racing striped cars were not "edge-case enough" for the attack to go through despite being semantically
sound. We note that the choice of "Southwest airplanes", however, is sufficient for a successful attack. (ii) *Construction*
*of p-edge case attack:* Yes, choosing a dataset that is entirely out of distribution *e.g.*, CIFAR in the MNIST example
would work. However, people are unlikely to use a model trained on MNIST for prediction on CIFAR examples. Thus
such a "fully out-of-distribution backdoor" is not as practical. We agree that constructing a *good p*-edge-case dataset is
non-trivial. We are actively working towards this direction. (iii) *Setup of $\mathcal{D}_{edge}$:* Yes, the default setting is that only the
adversary has access to $D_{\text{edge}}$. The absolute number of edge-case examples varies across experiments *e.g.*, for Task
1, we randomly sample 66 of the ARDIS images and mix them with 100 randomly sampled EMNIST images. More
details are included in Appendix A. (iv) *Evaluation:* Our evaluation did not include a comparison with BadNets since
the latter requires both training-time and inference-time access to the data for inserting and triggering the pixel-pattern
backdoor, while in the general FL scenario, the attacker has (at most) only training time access. We have added a more
comprehensive discussion on trigger-based backdoors including BadNets in the revision for completeness.
**R2**: *"Concerns related to Proposition 1 and 2."*
We have added more intuition in the main text as per your suggestion. Proposition 2 uses a simple construction to
demonstrate that there exist backdoors which are hard to detect. For scenarios having non-uniform data distribution, the
hardness of backdoor detection can be confirmed by Proposition 1.
**R2**: *"Clarification on the experimental results."*
(i) *Questions on Fig 4:* We partition the correctly labeled instances among a set of sampled honest clients while having
the attacker hold **all** incorrectly labeled instances. For Fig 4(a), we have 200 correctly labeled "Southwest" images. We
assign 100 to the attacker and 100 to 5 out of the 200 honest clients (each client holds 20). The goal is to create an
"almost edge-case" scenario where it is unlikely that the backdoor gets erased by clients who hold correctly labeled edge
examples. (ii) *Questions on the DP defense:* We agree that the NDC defense minimizes the influence of a particular
user's data. We did observe that NDC with a stricter norm bound leads to poor target and main task accuracy (sometimes
preventing full model convergence). Thus in our experiment, we tune NDC (and other defenses) such that the main task
accuracy reaches around $90\%$ in 500 FL rounds (for the EMNIST experiment in Fig 5(d)). When studying the weak
DP defense (built on top of NDC) under a wide range of noise levels, we observe that while high noise is effective in
defending against the attack, it hurts the main task accuracy too much (Fig 6(c)).
**R4**: *"Thm 1: $XX^{\top}$ being invertible – I am not sure if this assumption holds in practice."*
We verified this assumption by considering a FC ReLU network of width 2000 trained on MNIST. Randomly selecting
1000 data points gives us $\mathbf{X} \in \mathcal{R}^{1000 \times 784}$ of rank 574. However, the activation matrix of the first layer is $\mathbf{X}_1 \in$
$\mathcal{R}^{1000 \times 2000}$ which has rank $1000 \Rightarrow \mathbf{X}_1 \mathbf{X}_1^T$ is invertible. We have included the experimental results in the revision.
**R4**: *"[...] $\mathcal{D}_{edge}$ sets [...] actually having "exponentially small measure"? [...] heuristic [...] creating defenses?"*
The log density plot in Fig2 shows that ARDIS is of very small measure with respect to MNIST. We also included this
metric for other datasets in the revision, however the success of the attack further supports the claim. For practitioners,
a good heuristic would be the main-task accuracy of the model. Assuming that the test set is reasonably representative
of the underlying distribution, having uncompromised test accuracy proves that the backdoor is sufficiently small.

[Meta-Review · NeurIPS 2020]

The focus of the submission is backdoor attacks in federated learning. The authors 1) show that models prone to adversarial corruptions are also vulnerable to backdoor attacks, 2) prove that detecting backdoors can be hard, and 3) propose a new class of backdoor attacks called edge-case backdoors. The theoretical contributions are accompanied with extensive evaluation of the new backdoor attack on challenging datasets. The paper is technically sound, it focuses on a current topic of machine learning and delivers both important theoretical insights and new algorithmic tools. It can be of interest to the NeurIPS audience.